# Energy, Trophic Dynamics and Ecological Discounting

**Georgios Karakatsanis** [1,2,*] and **Nikos Mamassis** [1]

1   Department of Water Resources and Environmental Engineering, School of Civil Engineering, National Technical University of Athens (NTUA), 9 Heroon Polytechneiou St., 15870 Zografou, Greece; nikos@hydro.ntua.gr
2   Department of Research, EVOTROPIA Ecological Finance Architectures Private Company (P.C.), 190 Syngrou Avenue, 17671 Kallithea, Greece
*   Correspondence: georgios@hydro.ntua.gr; Tel.: +30-69-4555-2243

**Abstract:** Ecosystems provide humanity with a wide variety and high economic value-added services, from biomass structuring to genetic information, pollutants' decomposition, water purification and climate regulation. The foundation of *ecosystem services* is the *Eltonian Pyramid*, where via prey–predator relationships, energy metabolism and biomass building take place. In the context of existing ecosystem services classification and valuation methods (e.g., CICES, MEA, TEEB), financial investments in ecosystem services essentially address the conservation of trophic pyramids. Our work's main target is to investigate how trophic pyramids' dynamics (stability or instability) impact the long-run *discounting* of financial investments on ecosystem services' value. Specifically, a trophic pyramid with highly fluctuating populations generates higher *risks* for the production of ecosystem services, hence for ecological finance instruments coupled to them, due to higher temporal *uncertainty* or *information entropy* that should be incorporated into their discount rates. As this uncertainty affects negatively the *net present value* (NPV) of financial capital on ecosystem services, we argue that the minimization of biomass fluctuations in trophic pyramids via population control should be among the priorities of ecosystem management practices. To substantiate our hypothesis, we construct a *logistic predation* model, which is consistent with the Eltonian Pyramid's ecological energetics. As the logistic predator model's parameters determine the tropic pyramid's dynamics and uncertainty, we develop an *adjusted Shannon entropy* index ($H(N)_{ADJ}$) to measure this effect as part of the discount rate. Indicatively, we perform a Monte Carlo simulation of a pyramid with intrinsic growth parameter values that yield oscillating population sizes. Finally, we discuss, from an ecological energetics standpoint, issues of competition and diversity in trophic pyramids, as special dimensions and extensions of our analytical framework.

**Keywords:** ecosystem services; Eltonian Pyramid; discounting; risk; uncertainty; information entropy; net present value (NPV); logistic predation; adjusted Shannon entropy

## 1. Introduction

The massive and global-scale impact of human-induced environmental stressors has reduced the *primary biomass productivity* of global, regional and national ecosystems [1]. This negative effect is synopsized in the increasing *unpredictability* of ecosystem *functions* and *growth patterns* that deviate from the established periodicity of interlocked biogeochemical cycles, in the context of widely accepted classification frameworks of *ecosystem services* [2–4] as *final ecological goods that provide humanity with economic value added*, although they do not strictly belong to the anthropogenic economic production nexus. The concept of ecosystem services is relatively new in the natural resource economics literature, substantiating the economic value that ecosystem functions provide to the human economy [5] that would be either impossible or extremely expensive to be substituted by human technology [6]. Thus, a property of ecosystem services is their *non-substitutability* or *complementarity*. According to CICES 5.1, TEEB and MEA classification frameworks, as the most widely

accepted [2–4], ecosystem services concern both continental and marine landscapes, with their types ranging from biomass for nutritional (e.g., fish) and manufacturing purposes (e.g., construction, furniture), climate regulation, $CO_2$ absorption and metabolism, genetic information for medical research, water purification and flood protection to leisure, culture and tourism. Services in an ecosystem are essentially produced by biological populations that interact via prey–predator relationships as a pivotal part of the *Eltonian Pyramid* [7], or simply *trophic pyramid*, which has been established as the core conceptual construct of *ecological energetics*. The species in the trophic pyramid's levels, in turn, regulate the abiotic ecosystem elements, such as temperature, precipitation, evapotranspiration and the flow of nutrients (e.g., carbon, nitrogen, phosphorus, water).

The increasing scientific understanding of the contribution of ecosystem services to the human economy attracts respective amounts of financial capital, with *ecological finance instruments* as investment vehicles [8–10]. The main target of our work is to examine *how the uncertainty in trophic pyramids' populations and biomass production patterns affects the value of such investments in terms of long-run discounting* and construct a mathematical framework that is able to explain this mechanism, in consistence with the principles of *ecological energetics*. In this view, the trophic pyramid is the hierarchical structure of prey–predator relationships in an ecosystem [11], essentially regulating the flow and storage of energy in the ecosystem, in biomass form [12]. As the *objective function* of ecosystem populations is to maximize their energy flow and storage under constraints (e.g., nutrient scarcity for plants, competition for animals) [13,14], the trophic pyramid's energy flow and storage efficiency (thus, of the deriving ecosystem services) highly depends on the long-run stability of its constituent populations. In short, the *uncertainty* of the energy flow in trophic pyramids affects directly the uncertainty of produced ecosystem services, hence the associated *discounting risks* of financial investments on ecosystems' conservation.

Specifically, in order to examine our hypothesis and depict the above ecodynamics, we follow a sequence of specific steps from the conceptual foundations of logistic growth models. Our *first* step is to examine the features of continuous-time and discrete-time logistic population models and justify our preference for the latter. In this context, in turn, we re-postulate the concept of *carrying capacity* as an emerging biophysical property of the *limiting factor* [15]. After clarifying the conceptual background, we structure a simple population growth model that is based on the dynamics of the *logistic cobweb map*. To be consistent with the *Eltonian Pyramid*, populations are expressed in *energy-equivalent individuals* (EEPs) [15]. Preserving a trophic pyramid structure, we essentially build a *logistic predator* model [11,16], integrating elements from both classical prey–predator [17,18] and logistic growth approaches [19] and benefiting from the accuracy of the former and the simplicity of the latter [20]. Essentially, our approach allows the building of multi-level logistic predator models [21], where each level $EEP_{n-1}$ is practically the *carrying capacity* of the level $EEP_n$ of the trophic pyramid [22]. Plant populations are found at the base of the pyramid, where their carrying capacity is determined by the growth factor (e.g., radiation, water, chemical compound) in minimum relative availability or the *limiting factor*. The plants' limiting factor determines fundamentally the growth and size of the trophic pyramid. The higher the number of a trophic pyramid's levels, the higher its ability to regulate and store energy from the environment in the form of biomass.

After the formulation of the trophic pyramid as a dynamic logistic predator model, the *second* step of our work is to examine the mathematical properties of the model's *endogenous stability* [23], as this is a determinant of uncertainty production that is embodied in ecological discounting. In logistic cobweb models, the value of the *intrinsic growth parameter* is of definitive importance for a population's *stability* and *sustainability* [15,19]. We show that excessive *intrinsic growth rates* set the condition for unstable and unsustainable populations. To measure the impact of this instability per case, we postulate an *adjusted Shannon entropy* index [24] that is incorporated as the *risk factor* in the risk-adjusted *discount rate* of an ecological investment's *net present value* (NPV). Although logistic predator models are epistemologically deterministic, they can generate *stochasticity*. Based on this property,

we show that, essencially, the self-stabilization of an endogenously unstable (i.e., with a high intrinsic growth rate) population is *not impossible but highly improbable*. This property allows us to demonstrate mathematically that *exogenous* population control measures (e.g., harvesting, hunting) are able to restore the long-term stability of trophic pyramids, as studies in the field suggest [25–27]. The exogenous stabilization of populations reduces both the generated uncertainty and the risk factor in ecological discounting. For this part, we present the mathematical conditions for population stabilization in the logistic predator model and further perform a numerical simulation with five (5) trophic pyramid levels, with plants as *Producers* at its base and four (4) types of *Consumers* above it, as presented in Appendix A, to back our approach.

## 2. Materials and Methods

In this section, we examine the fundamental mathematical properties of continuous and discrete-time logistic growth models and re-postulate the concept of *carrying capacity* (as a critical parameter in logistic growth models) as an emerging property of the *limiting factor*. We also discuss the theoretical background of the concept of the *Eltonian Pyramid* and present global empirical evidence of its validity at the taxonomy level. Upon setting the background, we structure our multi-level logistic predator model. Finally, we briefly review the CICES 5.1, TEEB and MEA classification and valuation frameworks of ecosystem services and present literature evidence on the global importance of biomass for their production, as well as its high monetary value for the human economy.

### 2.1. Logistic Growth in Continuous Time

The majority of ecosystem population growth models can be classified into three (3) categories: (a) logistic growth, (b) prey–predator and (c) ratio-dependent. Each category focuses on specific ecosystem attributes. The first category is mainly expressed by Verhulst-type models and usually focuses on the growth of a single species with limited resources [19, 28,29]. The second category is mainly expressed by Lotka–Volterra models [17] and focuses on the co-evolutionary population size dynamics between two species related by a prey–predator relationship. The third category is mainly expressed by the Michaelis–Menten equation that was originally developed for modeling enzyme kinetics and further evolved by Holling (as type I and II) for modeling population growth dynamics based on prey and predator population densities [16]. Furthermore, every model category embodies specific properties regarding its endogenous characteristics, such as *competition*. A typical case of a ratio-dependent or density-dependent model with a rich variety of behaviors by its parameter values is the Hassel family [15,30], although even logistic growth models in discretetime can be re-formulated as density-dependent models [19], while overlapping between the above categories, is also possible.

Any ecological model should reflect a well-clarified hierarchy of principles, from the most fundamental one to the most optional [20]. It would be quite accurate to claim that the fundamental ecological principle—in line with the *second thermodynamic law*—dictates that no species can grow infinitely due to limited resources. For continuous time, we are concerned with modeling the *map* of population sizes $N$ across an exogenous time $t$, for a given *intrinsic growth rate* $r$ and an upper *maximum population capacity* $K$. According to this rationale, the logistic growth differential equation is:

$$\frac{dN}{dt} = r \cdot N \cdot \left(1 - \frac{N}{K}\right), \quad N \leq K, r, N, K \in R^+ \tag{1}$$

In Equation (1), the carrying capacity $K$ emerges physically from the concept of the *limiting factor* [31] in direct relation to the natural availability of nutrient resources. The limiting factor is the essence of the *Sprengel–Liebig's Law of the Minimum* for agricultural ecosystems, dictating that a system's overall growth will be determined by the resource found in minimum natural availability. Formulating the conditions for the existence of a limiting factor, we consider that from a set of resources $n$ combined by a system to form its

internal structure, for a specific and constant demand of a resource species $X_i$ found at a specific natural availability $Y_i$, the limiting factor determining the carrying capacity is:

$$K = Max[(X_1/Y_1);(X_2/Y_2);\ldots;(X_n/Y_n)], \quad X \leq Y, X, Y_{1 \to n} \in R^+ \tag{2}$$

In Equation (2), the carrying capacity $K$ of logistic growth models is simply defined by the resource in maximum *relative scarcity* (demand/stock); providing significant conceptual conveniences by requiring the identification of only the *least* available resource instead of the composition of a complex scalar carrying capacity index. According to the above, by integration, the solution of Equation (1) is:

$$N(t) = \frac{K}{1 + \left(\frac{K-N_0}{N_0}\right) \cdot e^{r \cdot t}} \tag{3}$$

As with every population growth model, continuous-time logistic growth models embody specific strengths and weaknesses. For instance, the above Verhulst-type model is unable to reproduce population size fluctuations or collapses, as Equation (3) always reaches a stable equilibrium for population size $N = K$, irrespective of the intrinsic growth rate value $r$. This feature is due to the fact that the model is a function of time as an exogenous variable, in contrast to discrete-time logistic growth models that are a function of population sizes at previous time steps. Hence, although a continuous-time model is useful to depict accurately numerous other phenomena (e.g., technological transitions), it has an endogenous weakness in regard to the depiction of the quantitative impact of an excessively rapid population growth rate. Additionally, within a trophic pyramid context, such a model would achieve a multi-level stable equilibrium that comprises only a special case, as it would neglect that trophic pyramid collapses could occur due to the elimination of species positioned near the base of the pyramid (e.g., plants).

As Lotka–Volterra models reflect the trophic relationships between two species, by not assuming any kind of growth limitation, they result in linear infinite prey growth in the absence of a predator [11] or in infinite periodical fluctuations for at least two species ($n \geq 2$) that are never stabilizing (although these fluctuations can be minimal). Finally, ratio-dependent models resolve the problem of different reproduction and predation time scales between two (prey–predator) species [11], however, by incorporating high functional difficulty when used for modeling interactions of at least three species ($n \geq 3$). Syntheses oriented to eliminate the above weaknesses have appeared in the literature, although the Verhulst population model was dominant until the first quarter of the 20th century. The postulation of Lotka's principles on prey–predator interactions [13,14], followed by Volterra's mathematical formulation [17], offered a first differentiation. Another differentiation came after the formulation of Michaelis–Menten–Holling equations [16]. Most subsequent models were developed as variations of these three types. P.H. Leslie achieved the first significant synthesis, by unifying logistic growth and prey–predator dynamics [32], essentially formulating the first *logistic predator* model.

### 2.2. Logistic Growth in Discrete Time

In this part, we examine theoretically the mathematical and graphical features of the *logistic cobweb map* as the simplest expression of discrete-time logistic growth models. The general mathematical expression of any discrete-time model is:

$$N_t = f(N_{t-1}), \quad N \in R^+, t \in N^+ \tag{4}$$

Equation (4) suggests that a discrete-time model essentially produces an iteration of a causal self-feeding sequence. In this case, the population at any time step $t$ will define the population size at time step $t + 1$. The outcome of this process (net growth, stability, net decrease or collapse) depends exclusively on the parameters' values. As for continuous

time, the most parsimonious logistic growth models consist of an *intrinsic growth rate* and a *carrying capacity* parameter. We may reformulate Equation (4) as:

$$N_t = r \cdot N_{t-1} \cdot \left(1 - \frac{b}{r} \cdot N_{t-1}\right), \quad r > b, r, b \in R^+ \tag{5}$$

In Equation (5), parameter *r* depicts the *intrinsic growth rate*, and parameter *b* depicts a *resource efficiency coefficient* or a *population limitation intensity coefficient* that derives from the impact of consuming the carrying capacity. Specifically, any individual for any positive initial total population *N* will grow at a rate of *r*, as the average number of offspring per individual. This consumption impacts the gross population growth at the next reproduction time step negatively by a *b* coefficient. This means that although the average intrinsic growth rate remains constant (=*r*), the population has an intrinsic tendency to reduce its gross growth rate due to the consumption of carrying capacity. This may occur via a higher number of deaths in the population or via intense competition and selection of offspring protected by the parents. To make Equation (5) consistent with standard *r-K* models, we set *b/r = 1/K*, so that carrying capacity *K* is the ratio of parameters *r/b* as:

$$K = r/b, \quad K \in (1, +\infty) \tag{6}$$

In Equation (6), parameter *b* expresses the *intensity* of *gross* population reduction in relation to parameter *r*, so that a temporary *net exponential population growth* is not a prohibitive case, although in the long run, the population will stabilize, converge or oscillate around an upper maximum value due to the prevalence of the limiting factor, as in Equation (2). The most typical patterns of logistic growth for discrete-timetime models are presented in Figure 1.

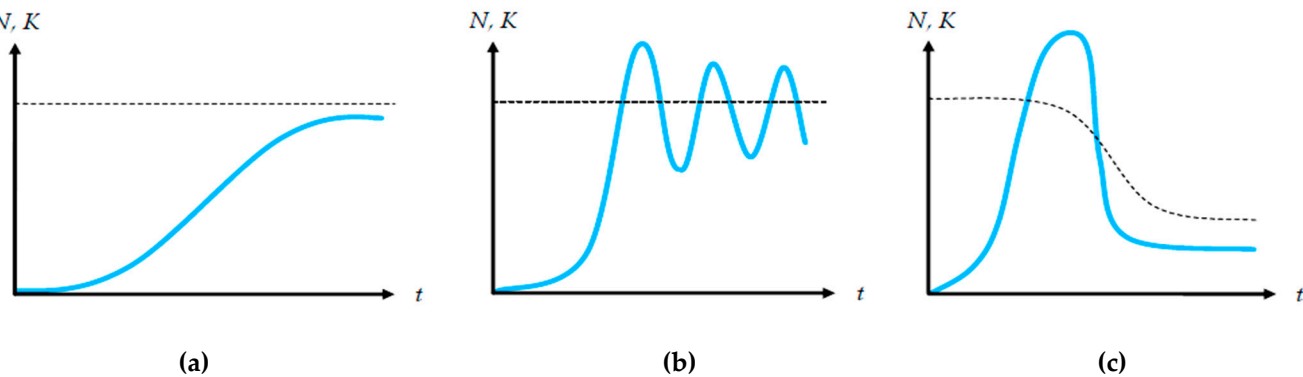

(a)    (b)    (c)

**Figure 1.** Schematic depiction of three main discrete-time logistic growth patterns with a carrying capacity (dashed line) constraint: (**a**) Globally stable convergence of the population to its carrying capacity. This pattern in most cases represents plant growth and is the unique pattern in continuous-time logistic growth models as in Equation (3). (**b**) Oscillating population size with a slow asymptotic convergence to its carrying capacity. This pattern could describe herbivores that overgraze their area, depriving it of its renewable elements periodically, resulting in their own population fluctuations. (**c**) Population growth overshoots its carrying capacity and reduces it permanently. Equilibrium in the system is re-established for a much lower population. This pattern could describe human-induced pollution, from land contamination from discharge of toxic wastes, land erosion and desertification from aquifer salinization or pesticide overuse.

Except for its ability to include a variety of growth paths (other than those of Figure 1a), a major mathematical aspect of Equation (5) is its potential use either as a *difference* or as a *growth* equation. In our current work, we utilize Equation (5) and its logistic predator reformulations as a *growth* function. Figure 2 depicts two growth paths of Equation (5), where population stability depends exclusively on parameter *r*, while the population size depends on the ratio *r/b*. By solving the continuous map *rN-bN²* as depicted in Equation

(5) for its maximization value $r/2b$ (from its first derivative), we find that the maximum stable population with no oscillations is for $r = 2$, at a size equal to $1/b$.

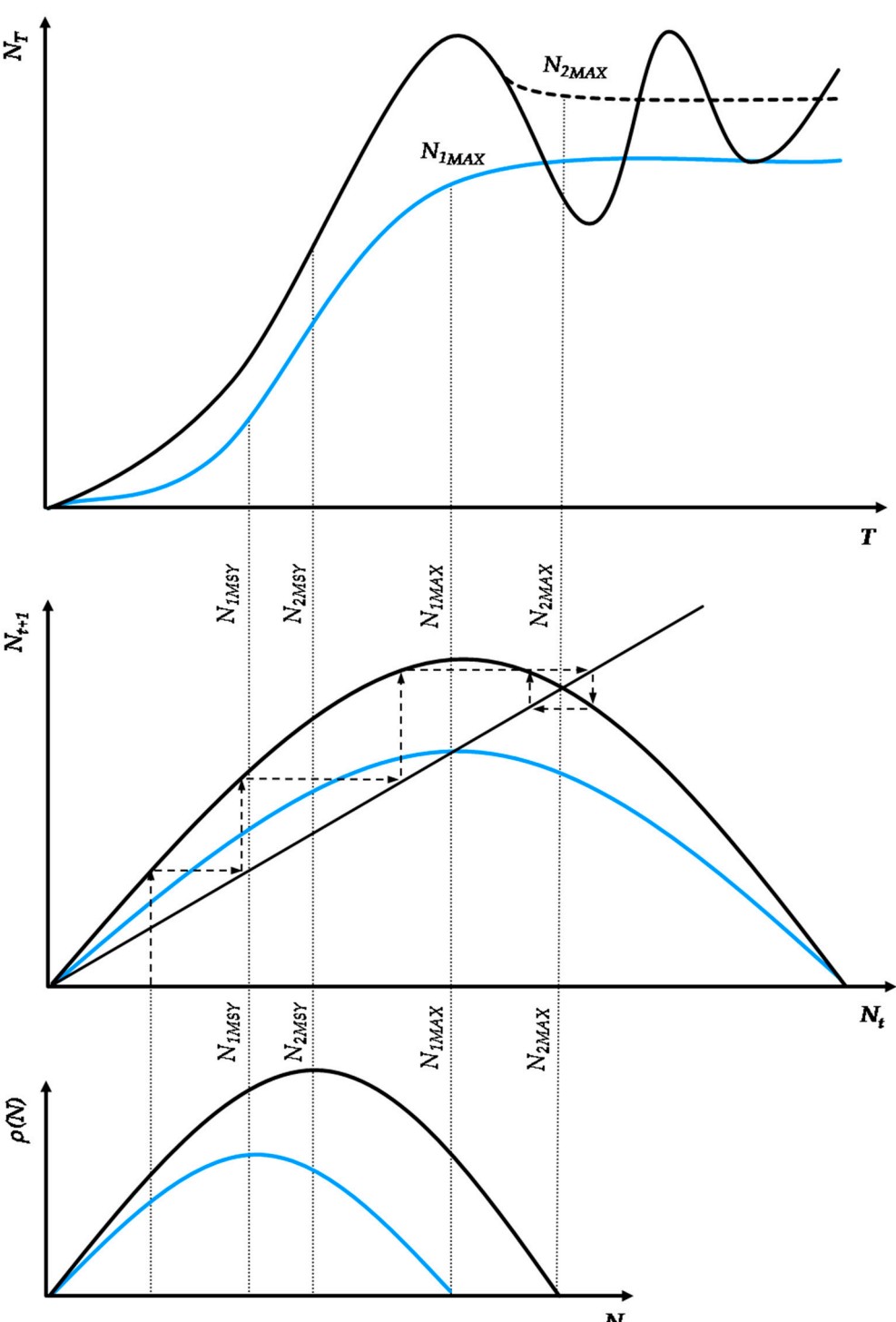

**Figure 2.** Schematic depiction of the *logistic cobweb map* with two growth paths: *optimally stable* (blue) and *convergent stable* (black).

The logistic cobweb map, as described in Equation (5), consists of three (3) structural elements: (a) the *growth rate plot*, (b) the *cobweb plot* and (c) the *temporal population growth plot*. In Figure 2, the bottom graph depicts the *growth rate map* $\rho(N)$, where its maximum value gives the *maximum sustainable yield* (MSY). At the *MSY*, the population grows at its maximum rate. The $N_{MSY}$ size is frequently set as a harvesting target so that the population

remains sustainable and at its maximum yield. The cobweb map in the middle derives from Equation (4) and depicts the *maximum stable population* growth path for *r* = 2 and an oscillating growth path for *r* > 2 as shown in the upper temporal $N_T$ size graph.

### 2.3. Energy and the Trophic Pyramid

A logistic predator model should typically comply with the fundamental notion of the *Eltonian Pyramid*, originally postulated by the animal ecologist C.S. Elton in 1927 and further specialized by Lindeman [7,33]. According to this concept, tissues store chemical energy. When that energy is released by trophic processes (such as prey consumption), most of it degrades into heat (as the *second law of thermodynamics* dictates). The remaining energy is insufficient to sustain the same biomass amount at the next trophic level. It was estimated by Lindeman [7] that ~90% of each energy flow across predation degrades into heat, due to body movement (e.g., running, hunting) and metabolism (e.g., defecation, reproduction), while only 10% is embodied in tissues for biomass formation and maintenance. Figure 3 depicts the trophic pyramid. With 10% energy efficiency, a large prey population can only sustain a lower predator population. A critical detail is that while the number of individuals is lower across the pyramid's ascension, the embodied energy per individual is higher. Figure 4 presents an empirical evidence of the above aspects [34].

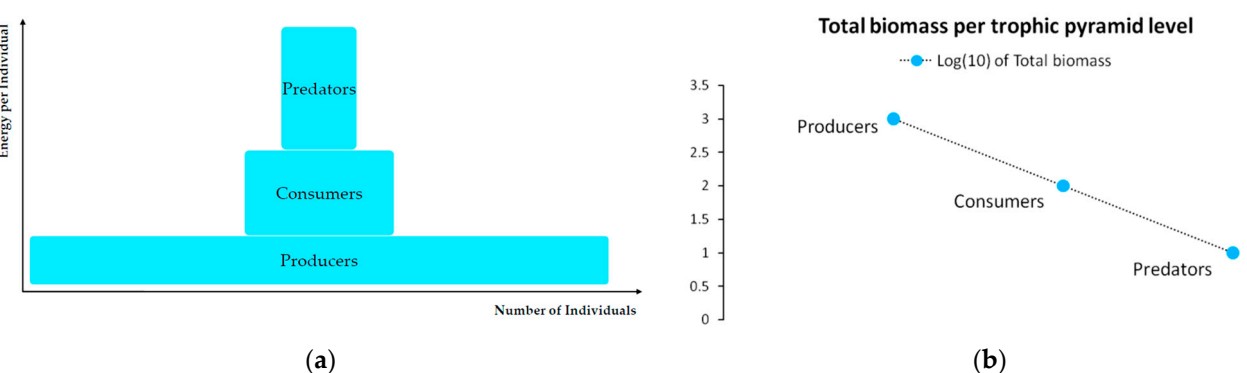

(**a**)          (**b**)

**Figure 3.** Schematic and quantitative depiction of Lindeman's pyramid: (**a**) A 3-level pyramid with *Producers* and two *Consumer* types; (**b**) Lindeman's total biomass (*B*) pyramid in $\text{Log}_{10}(B)$ terms.

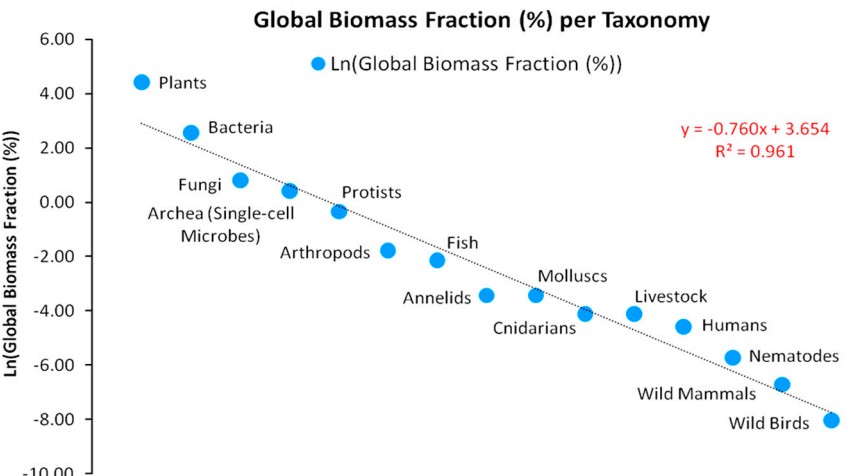

**Figure 4.** Global biomass fraction (%) per taxonomy of the total biomass forming the biosphere by global data [34] at natural logarithm (Ln) scale.

According to Figure 3b, a constant thermodynamic conversion coefficient of *h* = 0.1 is assumed across each flow of energy from a lower to an upper pyramid level. The biophysical meaning of such a conversion is that for 1 unit of prey biomass consumed by the

predator, only 10% will be metabolized into predator biomass. Hence, following Lindeman's estimations, the sustenance of 10 top predator individuals would require *at least* 100 available individuals of prey (primary consumers or herbivores) that in turn would require *at least* 1000 individuals of *Producers*. From this exponential relationship derives the linear relationship in the logarithmic scale of Figure 3b. In general, for each individual at a level *n* in the trophic pyramid, we can mathematically express Lindeman's rule for the minimum required individuals *N* at any level *m* < *n* in the trophic pyramid as:

$$N_{m \to n} = 10^{n+1-m}, \quad n > m, n, m \in N^+ \tag{7}$$

Equation (7) can be used for bidirectional energy flows, even to reduce the individuals of a lower pyramid level to individuals of a higher level. However, as physically the flow of energy occurs upward, we set the constraint *n* > *m*. In reality, trophic interactions include more elements than *Producers* and *Consumers* (with Consumers further classified into herbivores and predators), such as *Decomposers* that consist of micro-organisms that decompose residual dead biomass matter and break it down into chemical elements and compounds, replenishing the pool of nutrients of the land. Via their role, decomposers mitigate the limiting factors of plants, so that a new cycle of energy flow in the pyramid begins. In addition, the Eltonian Pyramid's energetics are consistent with the ecological window of the *universal energy hierarchy* postulated by Odum [35,36] and with the various allometric laws originating from Kleiber [37] that have linked body mass and metabolic rate. Although such laws have been heavily disputed [38], many authors find empirical confirmation for both plants [39] and animals [40] for an exponent value equal to $\frac{3}{4}$. In any case, whether we adopt directly Lindeman's energy efficiency estimations or derive biomass transformation coefficients from metabolic power laws, the structural elements of the logistic predator model developed in the next part remain unaffected.

An empirical view of the Eltonian Pyramid is provided in Figure 4 on the shares of taxonomy's biomass to total global biomass [34]. Data on the Earth's global biomass pyramid show an impressive prevalence of *Producers* and *Decomposers*, whose biomass accounts for 99.6% of total biomass, while of animals accounts for only 0.4%. Furthermore, plants dominate with an 82.4% share. Bacteria follow plants with a 12.8% share, followed by fungi (2.2%) and single-cell microbes (1.5%). In contrast, the biomass of domesticated animals accounts for 0.016%, of humans for 0.01% and of wild mammals for just 0.0012%. These data show that *plants are the foundation* of the trophic pyramid, comprising a critical hub of metabolized solar energy, as assumed in our logistic predator model.

### 2.4. Trophic Pyramid Dynamics: Logistic Predation

As shown in Figure 2, the identification of conditions triggering oscillating population growth patterns becomes a vital issue, in order for population control and stabilization measures to take place on time. A major challenge in ecological modeling is a synthesis of the best attributes of existing models. This consists in a model that reproduces the overall growth dynamics in consistency with the Eltonian Pyramid. The non-linear reformulation of Equation (5) gives the discrete time cobweb map of a species' population growth. For *b*/*r* = 1/*K*, Equation (5) becomes:

$$N_t = r \cdot N_{t-1} \cdot \left(1 - \frac{N_{t-1}}{K}\right), \quad r > 1, K > r, N \in R^+ \tag{8}$$

In Equation (8), a population's growth depends on its size in the previous time step, which also depends on the one before it and so on. As we have already argued, logistic predator models are founded on the trophic pyramid notion; thus, they specify *carrying capacity as prey availability*. Trophic pyramid models have often been found to accurately reproduce real observations [7,32]. The actual innovation consists in the *compartmentalization* of the ecosystem's biomass, where compartments depict the pyramid's levels with energy flowing upward, from prey to predator. This is achieved via the introduction of a

multi-level population hierarchy, where prey biomass comprises predator carrying capacity. Hence, population growth at any pyramid level *n* becomes a function of population size in the previous period, as well as a function of prey availability. Specifically, for a Consumer at a trophic pyramid level *n* that seeks prey at any *lower* level *m*, we may write:

$$N_{n_t} = f(N_{m_t}, N_{n_{t-1}}), \quad N \in R^+, n > m, n, m, t \in N^+ \quad (9)$$

Equation (9) is the reformulation of Equation (4) and the generalization of population dynamics in logistic predator models for *n* trophic pyramid levels. The first specialization concerning Consumers is for reformulating Equation (5) as a *general logistic predation equation*:

$$N_{n_t} = r_n \cdot c_{n|m} \cdot N_{n_{t-1}} \cdot \left(1 - \frac{c_{n|m} \cdot N_{n_{t-1}}}{N_{m_t}}\right), \quad r_n \cdot c_{n|m} > 1, N_m \geq c_{n|m} \cdot N_n \quad (10)$$

Equation (10) describes the discrete-time logistic predator growth map, with *c* being a new parameter introduced in the model and *K* substituted by $N_m$, which comprises the food resource and carrying capacity of any *n*-level species. The subscript *n* concerns the level at which a species stands in the food chain; therefore, it can only take non-negative integer values. For instance, level 0 (*n* = 0) refers to producers, level 1 (*n* = 1) to herbivores, while level 2 (*n* = 2) to primary consumers. Similarly, it applies to secondary and tertiary consumers (levels 3 and 4) up to the level of the top predator (say level *k*), which stands at the top of the trophic pyramid. Leaving aside the possible complexity of trophic pyramids deriving from interactions between a Consumer and all trophic levels below it, assuming for simplicity that a Consumer seeks prey *only at the level below it*, we reformulate Equation (10) as:

$$N_{n_t} = r_n \cdot c_{n|n-1} \cdot N_{n_{t-1}} \cdot \left(1 - \frac{c_{n|n-1} \cdot N_{n_{t-1}}}{N_{n-1_t}}\right), \quad r_n \cdot c_{n|n-1} > 1, N_{n-1} \geq c_{n|n-1} \cdot N_n \quad (11)$$

Differentiating from Equation (5), *r* is the *net average* reproduction ratio of biomass per unit of species during an *ecosystem period*. An ecosystem period is complete only after the species with the longest reproduction period has completely reproduced its biomass. Reproduction periods are not the same for all species, as some species reproduce at very fast rates, while others at very low ones. The average reproduction ratio is estimated by reducing the reproduction ratio of any *n*-level species to the *fastest-reproducing species* reproduction ratio in the food chain. For instance, a top predator may need even six months to reproduce its biomass with a single offspring. In this period, the fastest-reproducing species (usually a bottom-level species) may need only half a month to reproduce, which is twice per month and twelve times per six months. Thus, the average reproduction rate of the top predator in terms of the Producer is 1/12 = 0.08 (*r* = 8%) biomass units per half a month. The selection of the population of reference (usually *Producer* or *Top Predator*) for reducing the reproduction rates of the other pyramid levels is an issue of mathematical convenience; however, homogenizing intrinsic growth rates is necessary for establishing a common time step in the discrete-time logistic predator model. In addition, the fractional values of *r* for Consumers can be biophysically interpreted as the period of pregnancy, which is consistent with the requirements for increased prey biomass until a new full predator individual is born.

Furthermore, coefficient $c_n$ is necessary to transform predator and prey populations in the same unit (biomass) in consistency to Equation (7). For instance, as Lindeman's rule dictates, with 10% metabolic efficiency, 100 individuals of prey cannot sustain a population of 90 predators just because the condition of $N_n/N_{n-1} < 1$ is preserved. The biomass of each candidate prey species is an important information, as for a specific level of a predator's energy needs, more individuals of a small-sized prey are required [27,41]. In the previous example with 100 individuals of prey, if 10 units of prey are required for a single

predator, then the maximum sustainable predator population can be only 10 individuals. The natural meaning of coefficient $c_n$ is that *it expresses predator individuals into comparable energy-equivalent individuals of prey*; thus preserving the *limiting factor* property. As a result, each time that energy flows from one pyramid level to another, it is capable of sustaining a smaller number of individuals. Thus, the fundamental relation between populations in the pyramid is:

$$N_0 > N_1 > N_2 > \ldots > N_{k-1} > N_k \tag{12}$$

In addition, for all populations not to diminish in time, prey populations must fully cover the food requirements of predator populations. Thus, it must stand:

$$N_0 > c_{1|0} \cdot N_1 > c_{2|1} \cdot N_2 > \ldots > c_{k-1|k-2} \cdot N_{k-1} > c_{k|k-1} \cdot N_k, \quad N_k \geq 2 \forall n > 0 \tag{13}$$

Equation (13) sets the constraint that the number of energy-equivalent individuals $c_n \cdot N_n$ at any trophic pyramid level $n$ must be lower than the number of physical individuals $N_{n-1}$ at the exact lower pyramid level $n-1$. With the *Producers* at its basis, the *Eltonian Pyramid* essentially comprises a *limiting factor* mechanism for *Consumers* in the upper levels, since it is the population of prey that defines the maximum population of predators. According to Lindeman [7], empirical studies in local ecosystems have shown that food chains are not expected to have more than five levels, which is also confirmed by Odum [34,35,42], of course, provided that at the top level, energy must be enough to sustain at least two units of top predators so that reproduction is possible. However, the available biomass of all Consumer levels has to be examined in relation to the *fundamental* population in the pyramid. This population, as shown in Figure 4 with global data, is *Producers*.

As global data [34] suggest, Producers comprise a trophic pyramid's primary energy storages. Their main differentiation from species in upper trophic pyramid levels is their ability to *transform directly solar energy* via *photosynthesis to chemical energy*. All other species are biologically designed to embody the chemical energy of their prey, even herbivores as primary Consumers that depend on Producers' availability. In contrast, producers' growth is limited by the physical availability of nutrients [15], where one is usually the limiting factor as described in Equation (2). Essentially, accumulated biomass in all levels above Producers is secondary solar energy, transformed by Producers in the form of biochemical bonds. Thus, the general formulation of Producers' population growth is:

$$N_{0_t} = f(N_{0_{t-1}}, a, K), \quad a, K \in R^+ \tag{14}$$

Parameters $a$ and $K$ comprise the analogs of parameters $c_n$ and $N_{n-1}$ respectively, with $K$ defined by Equation (2), however, maintaining a similar function as prey–predator relations in Equation (9). Specifically, parameter $K$ expresses the total availability (either as a material deposit or environmental capacity) of the identified limiting factor, while parameter $a$ depicts the demand for it per unit of producer biomass. Thus, the discrete-time growth map of producer populations is:

$$N_{0_t} = r_0 \cdot a_0 \cdot N_{0_{t-1}} \cdot \left(1 - \frac{a_0 \cdot N_{0_{t-1}}}{K}\right), \quad r_0 \cdot a_0 > 1, K \geq a_0 \cdot N_0 \tag{15}$$

Equation (15) suggests that a Producer's population size at any time step $t$ is a function of the population at the previous time step $t-1$. However, since Producers are located at the lowest trophic pyramid level, their biomass growth is constrained by the limiting factor, whether this is radiation, physical space, water, nitrogen, phosphorus or sulfur. The Producers' limiting factors are fundamental and definitive for a trophic pyramid's energy flow and storage ability, as all biomass will be built and accumulated at Consumer levels proportionally to the growth potential of Producers. This conclusion is based on global data [34] presented in Figure 4 and extends to the planetary scale with *net primary production* being the background for the production of high-value-added ecosystem services [6,43]. Furthermore, limiting factors may vary across different latitudes and Koppen

climate classifications. For instance, the most frequent limiting factor in desert ecosystems is water and occasionally sulfur, in tropical ecosystems, it is physical space, and in arctic ecosystems, it is temperature. In more temperate climate zones, the limiting factors may have seasonal variations, like nitrogen and phosphorus alternations in lakes.

Some further considerations on the logistic predator model concern the assumption of a *linear* topology of prey–predator interactions in the trophic pyramid. This suggests that a predator at level $n$ depends exclusively on the $n - 1$ level prey availability. In many real cases though, predators have numerous alternative prey sources to which they resort if their main pool of prey diminishes, suggesting *hierarchical mesh* topology. In response, we could assume that every level of the trophic pyramid consists of the weighted average of all taxonomically similar species with *partial* or *complete substitutability*, with predators at the exact upper level preying on them.

Another aspect of the logistic predator model concerns non-migrating populations, which is a behavior frequently experienced by species in real ecosystems when dealing with resource scarcity. In the logistic predator model, as prey populations in lower pyramid levels diminish, predators are unable to meet their minimum energy requirements to maintain their biomass and diminish as well. An argument against this feature would be that predators could simply migrate to a nearby ecosystem with higher prey availability. A possible response to this argument could be that the spatial dimension in the logistic predator model is indirectly assumed, so that migration is a possibiity. In short, the diminishing predator populations in the logistic predator model could just signify their spatial exit from the specific ecosystem's diminishing prey availability or the combined effect of *contest competition* [15,28–30] over limited resources and migration.

Finally, another originality of the logistic predator model is that it allows a dynamic behavior of the trophic pyramid itself. Specifically, it includes the case where levels of the trophic pyramid go extinct without the whole pyramid collapsing as well. For instance, if predator populations in the upper pyramid levels go extinct, all lower levels will remain intact. Enrichment behaviors are also possible. As shown in the simulation of Appendix A for a 5-level trophic pyramid, tertiary and quaternary consumers' populations remain at zero size until a critical size of secondary consumers (as prey) is formed that will allow the growth of tertiary and, in turn, quaternary consumers. In ecological terms, this could be interpreted as the migration of upper-level predators from other nearby ecosystems due to prey sufficiency, as shown in Equation (13). In addition, this enrichment signifies the growth of the whole pyramid's energy flow and biomass storage via the formation of new upper levels. While the process of new pyramid levels' formation is straightforward, for pyramid levels' elimination, we could suggest the emergence of *scramble competition* behaviors due to unstable population growth (for $r_n \cdot c_n > 2$) [15,29]. Although discrete-time models (e.g., the Ricker model) assume by default scramble competition behaviors with high resilience at low population sizes [30], in logistic growth and predator models, intense scramble competition ($r_n \cdot c_n \sim 4$) leads to population collapse [15].

### 2.5. Ecosystem Services Classification

Trophic pyramids are biophysical constructs by which energy, mass and information flows are allocated in ecosystems. These flows determine fundamentally the provision of high value-added *ecosystem services* for humanity as *surpluses* (i.e., products and functions that are utilizable by the human economy without the latter needing to tether any endogenous resources to produce them). The thorough examination of the proposed classification frameworks [44] that comprise the scientific basis for political and economic decision making [45,46] escapes the goal of our study. However, we consider it necessary to dedicate this part to present the background of established ecosystem services classification systems, as we utilize their rationale for the formulation of an integrated ecological NPV model. A pivotal aspect for the establishment of ecosystem services classification and valuation frameworks was the proposal for a manual of a consistent framework for *corporate* and *national environmental cost–benefit accounting*, aiming at an integrated *System of Economic*

*and Environmental Accounting* (SEEA) [47]. Although its most recent version was formed in 2012 (and officially published in 2014) via the collaboration of the EU and a number of international organizations subject to the UN [47], the discussion for establishing "green" corporate and national accounts can be traced back to 1992, across the Rio Summit and the postulation of the *Agenda 21*. Identifying the gravity of the global loss of genetic resources, the conservation and restoration of *biodiversity* became a pillar of the sustainability agenda, further affecting fundamentally the direction of all frameworks on ecosystem services' classification and valuation.

Currently, we may distinguish three ecosystem services classification frameworks that are dominant in the global literature and practice: (1) the *Millennium Ecosystem Assessment* (MEA), (2) the *Economics of Ecosystems and Biodiversity* (TEEB) and (3) the *Common International Classification of Ecosystem Services* (CICES), which is also the most updated and widely accepted (currently at its version 5.1) [2]. The popularization of the ecosystem services concept essentially became feasible at the beginning of the 21st century via the MEA framework, which can be considered as the oldest, establishing the background for further updates and proposals. The first draft of the SEEA [48] coincided with the MEA, which followed the trend by highlighting the importance of biodiversity for ecosystems' functions and human well-being [49]. The TEEB framework followed in 2010 [50], where its major diversification from the MEA can be identified in its attempt to introduce valuation methods [3] as basis for *Payments for Ecosystem Services* (PES). The CICES framework was essentially developed in parallel with the SEEA; however, it became known only after the publication of its version 4.3 in 2013 (the most widely used version until the latest 5.1 revision in 2018), which harmonized with the MEA approach as its benchmark reference [2]. The MEA and TEEB approaches generally classify ecosystem services into four categories: (a) *provisional*, including material resources such as timber and hunting, (b) *cultural*, including nature-based recreation services, such as outdoor sports, eco-tourism and educational or field science activities, (c) *regulatory*, including functions for controlling energy, mass and information flows in the environment, such as flood risk reduction, CO sequestration and crop pollination, and (d) *supporting*, including genetic resources that can be utilized for nutritional and pharmaceutical purposes. The main diversification of the CICES framework is that it integrates *regulatory* and *supporting* services into a single category as it accepts their common biophysical basis. The current trend suggests that the CICES 5.1 framework is gaining ground. In any case, all frameworks tend to converge in their approaches, as irrespective of the discussion on how ecosystem services should be classified and valued, all of them adopt the above-mentioned typology.

*2.6. Biomass Productivity and Ecosystem Services Value*

Following the review of the prevalent ecosystem services classification systems, we may highlight the importance of biotic resources for their provision. In relation to classification and valuation frameworks, various compatible metrics have been suggested to depict biomass primary productivity, as well as its distortion by human-induced environmental impacts. A widely used index is the *Human Appropriation of Net Primary Production* (HANPP) [51] that is directly related to the concept of the *Eltonian Pyramid* as it estimates *the scale that land conversion and biomass harvest (that is directly related to provisional ecosystem services) by humans distorts the availability of trophic (biomass) energy in ecosystems*. Studies report an average global HANPP value of 32% [51]. The *gross ecosystem product* (GEP) [52] is a more elaborate index and compatible with the SEEA and TEEB frameworks, as it includes the dimension of monetary valuation of biomass productivity. Respective studies validate the importance of forests, belonging to the category of plants as shown in Figure 4, both in terms of biomass and monetary value productivity in the EU, following the CICES framework [53]. Based on these studies, we may argue that as ecosystem services are part of a landscape's *natural capital* stock, *a trophic pyramid comprises a functional structure of stored environmental value* that can be *monetized* as an incentive to attract public and private investments for its *conservation* instead of its depletion. The ability of ecosystems to produce

valuable life-supporting and economic services primarily relies on the stability of trophic pyramid populations, determining their biogeochemical metabolic networks' ability.

In addition, ecosystem resilience, stability and productivity are proportional to biodiversity and the number of trophic pyramid levels [54]. As already demonstrated, *in logistic predator models, carrying capacity is expressed as prey density* [22]. In turn, the pyramid's overall stability depends on population stability at each trophic level, so that an ecosystem produces both high and stable amounts of biomass. Empirical studies [6] show that tropical forests and coral reefs containing rich biodiversity and multi-level trophic pyramids are the most indicative cases where biomass production is estimated to be highest. The average carbon productivity in tropical forests is estimated to reach 900 $g/m^2/y$, followed by Mediterranean and Central European forests with an average productivity of 580 $g/m^2/y$ [6].

The trophic pyramid stability is a condition for the regularity of biosphere functions [55] and ecosystem services of high value-added. Indeed, monetary value estimations of ecosystem services [6] in tropical areas (e.g., Central and South America and South East Asia) are consistent with empirical data on plants' carbon productivity [34] as Figure 4 suggests. Plants essentially are the fundamental climate regulators via the intake of atmospheric *greenhouse gas* (GHG) surpluses. In short, plant populations are exceptional mechanisms of transforming $CO_2$ accumulations in the atmosphere—that from an environmental accounting view is a *cost* for human societies—into biomass. This generates a positive feedback of higher atmospheric $CO_2$ absorption (until the plants' limiting factor starts dominating) that turns a previous cost into a *benefit* for human societies. As mentioned in Section 2.5, this natural process constitutes a *surplus value* for human societies. However, as the benefits of ecosystem functions (that human technology cannot substitute) have a measurable value for the economic system, an ecological finance mechanism for their *conservation* needs to be established, incorporating their endogenous properties, such as instability and biomass yield *uncertainty*.

## 3. Results

In this section, we present the results of the structured logistic predator model. Particularly, the four aspects presented are as follows: (a) The *limits of stability and fluctuations to a population's sustainability* for the logistic cobweb map as the mathematical foundation of the logistic predator model. Specifically, we examine via trigonometric and optimization methods the maximum instability that a logistic cobweb map can tolerate without becoming unsustainable at any time step *t* ($N_t = 0$); (b) The relation between *uncertainty* as *information entropy* and *natural capital*. Specifically, as biomass in the Eltonian Pyramid is a form of natural capital, we examine the impact of its instability on the long-term supply of ecosystem services and develop an *adjusted Shannon entropy* index $H(N)_{ADJ}$ that is incorporated as a *risk coefficient* in ecological investments' discounting; (c) The mathematical conditions for *stabilizing* the Eltonian Pyramid via external population control (e.g., harvesting, hunting) and its effect on the reduction inthe $H(N)_{ADJ}$ risk coefficient; (d) The *integrated NPV framework* for financial capital invested in ecosystem services.

### 3.1. System Stability and Sustainability

Overall, the logistic predator model preserves the property of single-variable discrete-time logistic growth models, as even for unstable growth dynamics $\forall r_n \cdot c_n \in (2,4)$—as shown in Figure 2—and for any *n* trophic pyramid level, *there exists at least one population growth path that leads to the maximum stable population size* that satisfies the mathematical condition $Nn_0 \rightarrow Nn_1 \rightarrow Nn_2 \rightarrow Nn_3 \rightarrow \ldots \rightarrow Nn_{MAX(S)}$. The stabilization of any *n* trophic pyramid level will set the *necessary* but not the *sufficient* conditions for the stabilization of the upper *n* + 1 level. Practically, this means that a stable population at any level *n* will function as a *constant carrying capacity* for the population at level *n* + 1. In this case, the logistic predator model manifests the same behavior as any single-variable discrete-time logistic growth

model, as in Equation (5). However, the *sufficient* condition for stable population growth in level $n + 1$ concerns the value of $r_n \cdot c_n {\sim} 2$.

Population oscillations in trophic pyramids occur either due to natural or anthropogenic pressures. Natural seasonal changes in nutrient availabilities usually cause stable small-scale cycles; therefore, they are predictable and seldom need any significant intervention [42]. Contrarily, uncontrolled large-scale unstable oscillations are quite sensitive to anthropogenic amplifications. Thus, in the context of the logistic cobweb map, external population control is an available option to prevent permanent population depletion of a trophic pyramid's level and the ecosystem value attached to it [6,56]. Many models have assessed this approach for logistic predator models [18], as the elimination of one trophic level will expand to all levels *above* it, eliminating a significant part of the food chain and leading the ecosystem to a lower biomass equilibrium. However, an emerging question within the analytical framework of the logistic cobweb map (whether the standard or the logistic predator) is to identify the *marginal conditions*, that being *the maximum instability that a population can tolerate without becoming unsustainable*.

Figure 5 combines Equation (6) with a trigonometric view of the instability limits of the logistic cobweb map. In trigonometric terms, for an angle ($a$) formed by the vertical lines $N_t = 0$ and $N_t = K/2$, with the hypotenuse connecting the points $(N_t, N_{t+1}) \rightarrow (0, r/2b)$, the *optimal-stable population* $N_{MAX}$ (capped) is reached for:

$$\overline{N_{MAX}} \rightarrow cos(a)/sin(a) = 1 \tag{16}$$

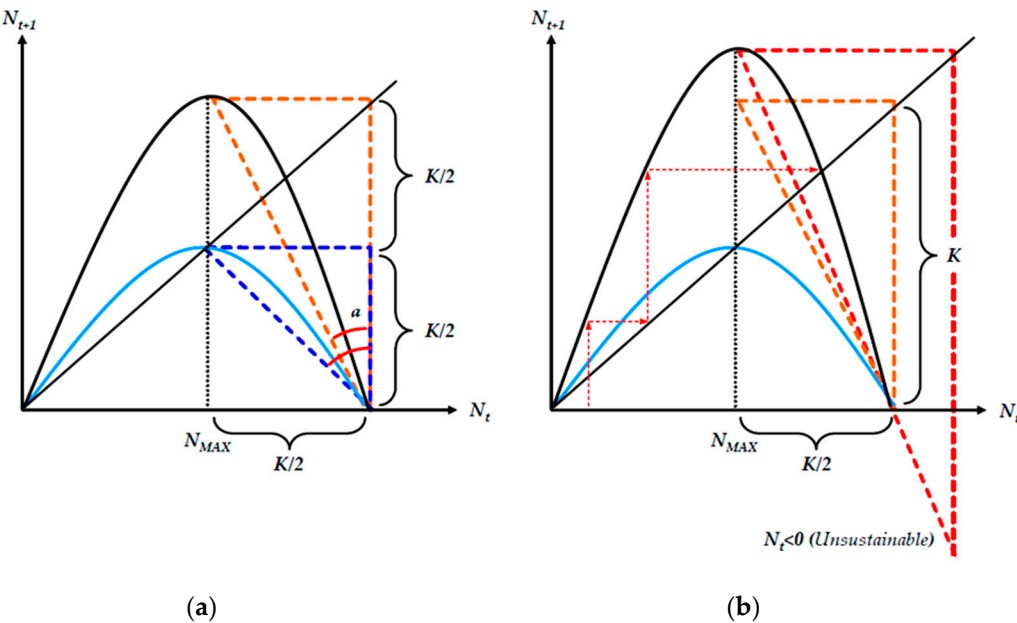

(**a**)　　　　　　　　　　　　　　　(**b**)

**Figure 5.** Graphic depiction of the conditions (in terms of *intrinsic growth rate* parameter values) that lead to *suboptimal-stable*, *optimal-stable*, *unstable-sustainable* and *unstable-unsustainable* population: (**a**) An optimal population size is achieved for $r = 2$ (blue map) that is stabilized at the maximum point of the logistic cobweb map without fluctuations at the $K/2 = r/2b$ point and at a size $1/b$. The system will stabilize at a suboptimal population size for any value $r \in (1,2)$, which is a stable population but at a size below the globally optimal ($K/2$). Both solutions are sustainable. For any value $r \in (2,3)$, the system will be unstable but convergent to a stable value, while for $r \in (3,4)$, it will be unstable with increasing fluctuations but asymptotically sustainable ($N > 0 \, \forall \, t$) (black map). (**b**) In contrast, for $r = 4$, $f(K/2 = 2/b)$ at time step $t$ yields population size $K$ at time step $t + 1$ that in turn gives an *unsustainable* ($N \leq 0$) population size at time step $t + 2$, as $t + 1$ size violates the upper sustainability limit $f(K/2) < K$.

Equation (16) sets an optimal intrinsic growth benchmark, where populations reach their maximum size without fluctuations. A value of $\cos(a)/\sin(a) \in (0,1)$ suggests that the population is stable but suboptimal, with maximum size below $1/b$. In contrast, for values $\cos(a)/\sin(a) \in (1,2)$ the population is unstable but sustainable. For values >2, the population becomes unsustainable. In Figure 5, the maximum intrinsic growth rate is expressed as at which $r$ value the function $f(K/2)$ at time step $t$ will yield exactly the value of $K(=r/b)$ at time step $t+1$ (forming the orange triangle). For $r = 2$, the formed triangle is an isosceles orthogonal (blue), giving a symmetrical population size where $N_t = N_{t+1} = K/2 = 1/b$.

We may further express the parameter's maximum $r$ value for a *globally sustainable* population (whether it is stable or unstable) in the logistic cobweb map, as a maximization function of the following formulation:

$$Max\left\{(r)\Big|\lim_{N \to K/2} f[f(K/2)] = K\right\} \tag{17}$$

Equation (17) seeks the maximum value of $r$ so that the $t+1$ iteration of Equation (5) for a value $N_t = K/2$ at $t$ gives a population reaching $K$. In turn, a population size $K$ at $t+1$, in the $t+2$ iteration, will asymptotically converge to zero $N_{t+2} = limf(N_{t+1}) \to 0$. The Lagrangian form of Equation (17), where parameter $\lambda$ stands for the *constraint multiplier* is written as:

$$L(r;\lambda) \equiv f(r) - \lambda \cdot g[f(f(K/2)) - K] \tag{18}$$

By solving Equation (18), we find that the upper limit of the *intrinsic growth rate* parameter value $r$ for an asymptotically sustainable population is $r = 4$.

Considering that the lower sustainability limit is for $r > 1$ (so that at least one value of the map is above the $N_t = N_{t+1}$ 45° line), we may conclude that for a constant carrying capacity value $K$, the sustainability conditions are met for any initial population size $N_0$ for $r \in (1,4)$. From Equations (6)–(8) and (15)–(18), as well as Figure 5, the following stands:

$$N_t \leq 0 \forall N_t = f(N_{t-1}) \geq K \tag{19}$$

However, as in logistic predator models, the carrying capacity may be variable for a population at a trophic level $n$ due to the fluctuations of the population at the level $n-1$, the optimization is not constant for every time step $t$. In the next part, we examine how population instability affects biomass supply *uncertainty* and develop a related *information entropy* index as part of the *risk-adjusted discounting* of ecological finance models.

### 3.2. Entropy and Natural Capital

The causal relation between *thermodynamic entropy* and *natural resource scarcity* has been examined in many classical essays on natural resource economics [57]. We generally extend such considerations to the relation between *information entropy* as a fundamental mathematical index of both classical thermodynamic entropy and natural capital supply *uncertainty*, as it can be applied in a straightforward manner even in renewable resources, such as biomass in a trophic pyramid. Our core argument is that *the value of natural capital is reverse proportional to the uncertainty of its provision* [58–60]. In short, *if biomass in a trophic pyramid is fluctuating and its size is highly uncertain across time, the provision of derived ecosystem services will suffer from respective uncertainty that will affect negatively their valuation*. In this context, we further identify endogenous financial risks, such as the higher insurance costs on ecosystems as carbon sinks [43], high discount rates deriving from the high uncertainty of forest biomass [61] and the increased premiums for insurance contracts on compensations regarding weather and climate operational risks [62]. Following this rationale, we develop an *adjusted Shannon entropy* index that is suitable for measuring uncertainty in trophic pyramid populations as a special case. Within this context, accurate ecosystem population modeling becomes an integral part of long-term sustainable ecosystem services conservation management as part of a society's total wealth [63].

The quadratic map as the basis of the discrete-time logistic model may demonstrate mathematical *non-invertibility* and *irreversibility* as its physical equivalent. As Mackey [64] proves, the continuous quadratic map is non-invertible, as for every population $N_{t+1}$, there are exactly two $N_t$ sizes generating every $N_{t+1}$ population size, except for the maximum point $r/2b$ that is unique and globally stable for $r = 2$. In Figure 6a, two different quadratic maps are presented; one reaching the optimally stable population for $r = 2$ (blue) and one with an unstable chaotic but marginally sustainable dynamics for $r = 3.99$ (black).

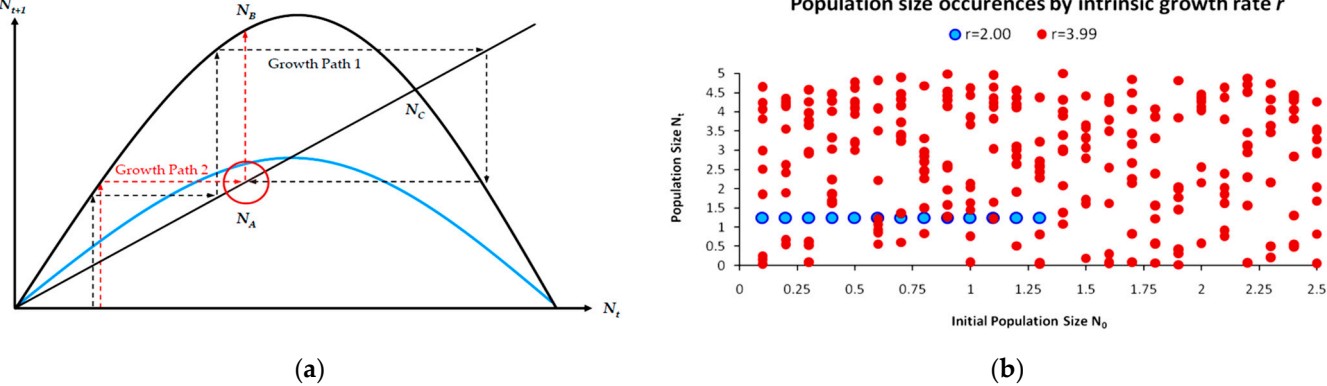

(**a**) (**b**)

**Figure 6.** Cobweb plot of two quadratic maps: (**a**) an optimally stable (blue) for $r = 2$ and a chaotic for $r = 3.9$ (black). While for the stable map, the arrival to point $N_A$ from any initial population $N_0$ comes from a *unique* growth sequence that can be exactly reproduced backward by reversing time, for the unstable map, reversing the growth sequence from $N_A$ generates at least two paths. (**b**) Population size occurrences for 10 iterations of Equation (5) for $b = 0.8$ for each quadratic map and 10 different initial population sizes $N_0 \in [0.1, f(r/2b)]$, counted from the first $N_t \geq r/2b$ and after $\forall N_0 \leq f(r/2b)$.

Discrete-time functions embedded in the continuous quadratic map manifest a very interesting behavior that depends on the value of parameter $r$. In Figure 6a, the growth function based on Equation (5) for parameter $r$ values $r \in (1,2]$ is completely reversible, as for every initial population value $N_0$, there is a *unique* growth path where each intermediate value lower than the maximum stable population appears only once; hence, if time is reversed, the system knows exactly which path to follow down to the starting point. In principle, a globally monotonic increasing function $f(N_{t-1}) \leq f(N_t) \forall N_{t-1} \leq N_t$ is *fully invertible and reversible* as it conserves the full memory of its growth path. As the value of parameter $r$ increases (for $r > 2$), the system becomes *partially invertible*, since the map begins to acquire more than one $N_t$ population size that yields a respective population $N_{t+1}$. In Figure 6a (black map), this occurs for the range of population sizes $N_t$ that satisfy the condition $f(N_t) > N_C$. The general condition of partial invertibility is a parameter value $r \in (2, \sim 3.6)$ that yields an oscillating population of stable periods (of 2,4 or 8 cycles). For $r > 3.6$, the system becomes *fully non-invertible* with such a high number of bifurcations that the population growth path becomes random (chaotic) for the major part of the map [19,65].

A better view of non-invertibility mechanics is presented in Figure 6a comparing the two maps for $r = 2$ and $r = 3.99$. Beginning from a random population size $N_A$ and reversing time by feeding backward Equation (4), for the optimally stable map with $r = 2$, there is a *unique* path leading to the initial population $N_0$ at $t = 0$ that is *exactly the same* as the one that the system followed across $N_0 \rightarrow N_A$. The repetition of this process infinitely will always yield the exact same result. Changing the random population size $N_A$ and reversing the process down to a defined $t = 0$ will always yield a unique path that is exactly the same as the one for a growing population. In contrast, for the $r = 3.99$ map that is the upper extreme case of oscillating but sustainable population size function (according to Equations (18) and (19)), the identical initial conditions for a random $N_A$ size would yield a different result. As shown in Figure 6a, there are *at least two* values that yield the population size

$N_A$—one generated by *Growth Path 1* and one by *Growth Path 2*—so that by attempting to reverse the process, the system would be unaware of the direction it should follow to go back to $N_0$. Alternatively stated, if we iterated for the $r = 3.99$ map the reversal process infinitely without any constraint, we would receive a distribution of two results with equal probability (=0.5). Thus, we would have (a) a reversal following *growth path 1* down to $N_0$ and (b) a reversal following *growth path 2* down to $N_0$, meaning that the system 50% of the time would choose a different path to go back to its initial state. This is not the case for the $r = 2$ system that will always follow the same path in reversed time.

In physical terms, the $r = 2$ condition in the optimal stable map essentially establishes a *maximum entropy* (MaxEnt) growth path with a sufficiently strict thermodynamic *constraint*. This constraint ensures that the possible population size sequences are only the ones that belong to the globally monotonic part of the continuous quadratic map. These population sequences are from an *information theory* (or *statistical mechanical*) perspective the system's *configurations* [66]. The constraint of the optimal stable map (blue) in Figure 6a essentially ensures that whatever the initial population $N_0$ and the growth sequence may be, it will converge to the maximum stable population $N_{(S)}$. In addition, each population path sequence is either *unique* $\forall N_0$ or a *subset* of another growth path (for instance, for two different growth paths $G_{(A)}$ and $G_{(B)}$, with $G_{(A)}$: $N_0 \to N_1 \to N_2 \to N_3 \to \ldots \to N_{(S)}$, if the initial population for $G_{(B)}$:$N_0 = G_{(A)}$:$N_1$, then $G_{(B)} \subseteq G_{(A)}$). This also applies to any suboptimal stable map for $r \in (1,2)$.

In contrast, as shown in Figure 6b, for the $r = 3.99$ map, the constraint is so weak that each population size at any time step $t$ can be derived by more than one population size at $t - 1$, forming a continuous field of values across infinite iterations. Inside this field, the system's growth path reversal can only be stochastic as the system has numerous options for returning to its initial state other than the same way it initially moved away from it. A stochastic view of Figure 6b is presented in Figure 7, where population sizes are generated with a *normal distribution* for indicative maps of $r = 2.00$, $r = 2.99$, $r = 3.40$ and $r = 3.99$.

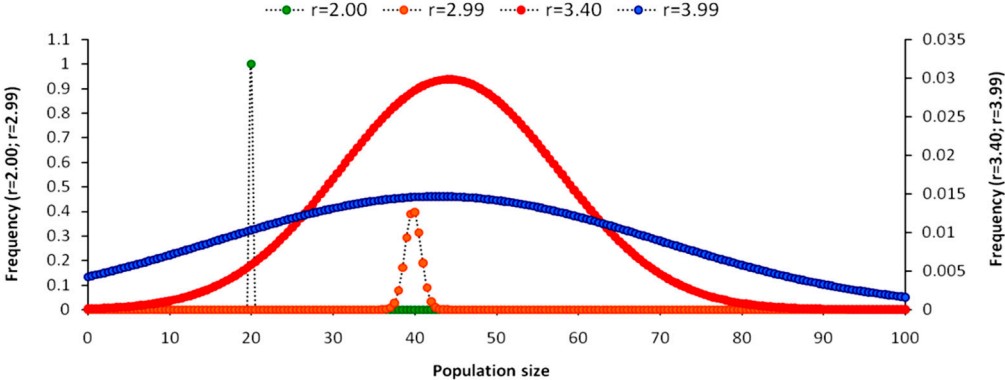

**Figure 7.** A stochastic approach of stability via generating population size values with the normal distribution from the first $N_t \leq r/2b$ value and after $\forall N_0 \leq f(r/2b)$ for 4 different $r$ parameter values.

Figure 7 presents the generated population sizes' range for 70 time steps $\forall N_t \geq r/2b$ for four different parameter $r$ values and a benchmark value $b = 0.05$, for which Equation (5) manifests different behaviors. Hence, for $r = 2$, the population size yields a constant population $N = 20$ with probability $p = 1$. For $r = 2.99$, which yields a convergent stable population $N \sim 40$, the probability becomes $p = 0.4$ with the remaining probability density distributed in the range between $N \sim (38,40)$ and $N \sim (40,41)$. As the parameter $r$ value increases, the distribution spreads out with increasing standard deviation and with a lower probability for each population size. For $r = 3.99$ (as the upper limit of a sustainable population according to Equations (18) and (19)), the probability density approaches asymptotically (i.e., for infinite iterations) *equiprobability* for all population sizes, suggesting

the emergence of the *uniform distribution* that expresses the unconstrained *universal maximum entropy* state. Quantitatively, we use the *Shannon entropy* $H(N)$ [24] to measure the required number of *nats* (the analog of *bits* for natural logarithm expressions) as:

$$H(N) = -\sum_{i=1}^{n} p_i \cdot \ln p_i, \quad t \in [(t_0 : r/2b \leq N_t), +\infty] \tag{20}$$

Typically, although for continuous distributions information entropy is proportional to the standard deviation ($H(N) \propto \sigma$), for empirical time series that derive from the various parameter $r$ values in Equation (5), the behavior can be contradictory. For instance, Equation (20) yields lower entropy for $r = 3.40$ than for $r = 2.99$ for a bin value ($=0.5$), as in the first case, the oscillation is wider but only between two different values due to the constant two-period cycle. To normalize Equation (20) and include the effect of the range of the $N$ size variability, we re-write:

$$H(N)_{ADJ} = \left( \frac{MaxN_{i|r_i} - MinN_{i|r_i}}{MaxN_{i|r_i}} \right) \cdot \left( -\sum_{i=1}^{n} p_i \cdot \ln p_i \right), \quad t \in [(t_0 : r/2b \leq N_t), +\infty] \tag{21}$$

Equation (21) adds to Equation (20) an *oscillator* as a *weight* to the Shannon entropy function to include the effect of range. The oscillator consists of the population's observed oscillation (i.e., between the population's maximum and minimum observed value) for a given value of parameter $r$ as a fraction of the maximum and minimum ($N = 0$) population values. This approach addresses the need for a *risk-adjusted* entropy measurement that is of high importance to financial investments in ecosystem services and the risk-adjusted *discount rates*. Specifically, an ecosystem may be considered to be at lower risk when oscillating randomly within a very narrow range and around its stable population size than an ecosystem with stable oscillation cycles but within a very wide range. Table 1 presents the measurements of both standard and adjusted Shannon entropy metrics for the four parameter $r$ values.

**Table 1.** Standard and adjusted Shannon entropy values of the observed population sizes at 70 time steps and for four different parameter $r$ values.

| $r$ | Behavior | (Max − Min)/Max | H(N) | H(N)$_{ADJ}$ |
|---|---|---|---|---|
| $r = 2.00$ | Stable | 0.0000 | 0.0000 | 0.0000 |
| $r = 2.99$ | Convergent Stable | 0.0686 | 1.6049 | 0.0110 |
| $r = 3.40$ | 2-period Oscillating | 0.4668 | 0.6930 | 0.3235 |
| $r = 3.99$ | Chaotic | 0.9885 | 3.9822 | 3.9364 |

The oscillator coefficient in Table 1 essentially comprises a system's entropy intensity index, showing within which range the system is random or uncertain. In this context, the $r = 3.99$ system is completely entropic as the values of standard and adjusted entropies are very close. The main changes are observed for the $r = 2.99$ and $r = 3.40$ as substantiated above. In contrast, the $r = 2$ system with zero entropy value for both metrics conserves its energy without ever deviating from $N = 20$ with $p = 1$ as the lower entropy limit ($=0$), which is also consistent with the physical expression of the *second thermodynamic law* ($\Delta S \geq 0$).

### 3.3. Stabilizing Trophic Pyramid Populations

Following the structure of the logistic predator model in Section 2.4, in Section 3.3, we present the mathematical conditions for stabilizing a population via optimal harvesting and growth control. A step-by-step analytical presentation of the logistic predator model and the mechanism of population stabilization in the Eltonian Pyramid are developed in Appendix A via a numerical simulation of a five-level trophic pyramid. The first

important aspect of population stabilization concerns the extension of its application. In principle, stabilization via population control should mostly concern highly unstable ecosystems [25,26], where it is expected to maximize ecosystem stability with minimum environmental impact [25].

A second emerging issue concerns the selection of the optimal trophic level for applying population control. Simply stated, it is of pivotal importance to identify which is the *key population* whose control will stabilize all levels above it in the Eltonian Pyramid. Most biological conservation approaches adopt as a common feature the identification of key species in the trophic pyramid that are both sensitive to changes of environmental parameters and important for ecosystem services. After assessing habitat requirements, the best option is adopted for enhancing the flow of energy and nutrients in the pyramid to sustain (and grow) its biomass formation [21]. More specifically, in a logistic predator model, it is necessary to have knowledge of the trophic level that is the cause of the oscillation. Targeted control of that level will mitigate (or even eliminate for a value of $r = 2$) oscillations in upper levels of the pyramid [21]. Generally, a recommended practice is for population control to follow a bottom-level approach, meaning that it must be applied to the first oscillating population closer to the trophic pyramid's base [21]. In logistic predator models, it is highly probable that populations oscillate due to the oscillations of the exact lower level at the role of their carrying capacity (prey), as a feature mostly inherited by prey–predator models. Except for top predators, every trophic pyramid level comprises the carrying capacity for its upper level; hence, if oscillations exist in the pyramid, there is a key population that initially generates them. If oscillations continue to occur to higher pyramid levels even after controlling the key population's size, the next oscillating population must be the next stabilization target and so on. In our simulation in Appendix A, this population is *Producers* (plants) at the foundation of the Eltonian Pyramid. Their control leads all *Consumer* levels (herbivores and carnivores above plants) in the pyramid to stabilize as well. In principle, the more fundamental is the level where the population control is applied, the higher is the probability to stabilize all other levels above it.

Population control consists in preventing excessive population growth in relation to carrying capacity. A population not perfectly adapted to its carrying capacity but oscillating around it can be stabilized via the control of its growth path. A facet of major importance for both logistic cobweb maps and the logistic predator model is that even for populations with unsustainable intrinsic growth rates $r > 4$, there exists *at least one possible growth path that leads to a long-term stable population $N_{(S)}$, $N_0 \rightarrow N_1 \rightarrow N_2 \rightarrow N_3 \rightarrow \ldots \rightarrow N_{(S)}$* [15]. For instance, even for the unsustainable map (black) in Figure 5b, there exists one optimal growth path (red dashed arrows) that begins from an initial population size $N_0$, leading to a perpetually stable population $N_{(S)}$. The critical feature here is that even for extremely high values of the intrinsic growth rate parameter value $r$, such a path exists for *any* map; however, *although it is not impossible, it is highly improbable*. As presented in Section 2.4, in globally stable maps $(1 < r \leq 2)$, *any initial population $N_0$* will lead to a stable population size. In contrast, for unsustainable maps $(r > 4)$, only a handful of growth paths out of an infinite pool will lead to stability. Hence, the spontaneous or emerging stability of unstable and unsustainable maps is an extremely rare phenomenon if infinite Monte Carlo repetitions are performed. Alternatively stated, for a map with $r > 4$, if we repeated infinite times its growth process, choosing every time a different initial population size $N_0$, we would receive unsustainable population sizes $N_t \leq 0$ with probability approaching *asymptotically $p = 1$*. In contrast, following the same process for any map with $1 < r \leq 2$ will yield perpetually stable population sizes with probability $p = 1$. This aspect is directly related to how *information entropy*, as a measure of uncertainty, impacts the supply of natural capital and its coupled ecosystem services, as well as how population stabilization as a mechanism for uncertainty reduction has a direct economic value on the NPV of ecological investments and their discounting, as we discuss in Section 3.4.

The link between intrinsic growth rates and population instability has been substantiated in May's original work [19,65] on the logistic cobweb map. In logistic predator models,

this may occur due to a combination of high $r$ and $c$ values. As in standard logistic cobweb maps, for each trophic level $n$, there is exactly one optimal population size $\overline{N}_n$ for which the population stabilizes perpetually for any parameter combination of $r$ and $c$, with energy constraints preserved. Especially for intense, near-chaotic population oscillations, external control may contribute significantly to the achievement of that target. The above applies at all trophic pyramid levels, from Producers to top predators. For a finite number of time steps and time frame $t - 1$ to $T$, when this optimal population size is reached, it stands for every ecosystem period that:

$$N_{n_{t-1}} = \overline{N}_{n_t} = N_{n_{t+1}} = \ldots = N_{n_{T-1}} = N_{n_T} \tag{22}$$

By combining Equation (9) with Equation (11), we may formulate the stable population size of a trophic level $n$ in the logistic predator model as:

$$\overline{N}_{n_t} = r_n \cdot c_{n|n-1} \cdot N_{n_{t-1}} \cdot \left(1 - \frac{c_{n|n-1} \cdot N_{n_{t-1}}}{N_{n-1_t}}\right) \tag{23}$$

By Equation (23), $\overline{N}_{n_t}$ is the desirable stable population size at the current ecosystem period, depending on the growth of the population size of the previous period. By Equation (23), determining the desirable population size for every period, it is easy to determine which population size at the exact previous time step $t - 1$ yields it. This applies to both the standard logistic cobweb and the logistic predator models, although the latter incorporates higher complexity. Therefore, we may conclude from the above that in order for a population $N_{nt - 1}$ to be equal to $\overline{N}_{n_t}$, it must generally stand that:

$$r_n \cdot c_{n|n-1} \cdot \left(1 - \frac{c_{n|n-1} \cdot N_{n_{t-1}}}{N_{n-1_t}}\right) = 1 \tag{24}$$

Equation (24) is a reformulation of Equation (23) by setting $\overline{N}_{n_t} = N_{nt-1}$ and dividing by the latter term, so that the population size at time step $t - 1$ is the only unknown value. Solving Equation (24) by $N_{nt-1}$, we have:

$$N_{n_{t-1}} = \frac{N_{n-1_t} \cdot \left(1 - r_n \cdot c_{n|n-1}\right)}{-r_n \cdot c^2_{n|n-1}} \tag{25}$$

Equation (25) is a second-order equation with $N_{nt-1}$ being the only unknown variable. According to the constraints of Equation (13), $1 - r_n \cdot c_n$ is always negative; therefore, Equation (25) yields only positive values, which is consistent with the model's constraints. An additional significant feature is that Equation (25) expresses the optimal population of an $n$-level species as a function of its carrying capacity (or a function of the $n - 1$ level species' available biomass). Alternatively, the optimal solutions may be found for every period through solving Equation (25) initially by $\overline{N}_{n_t}$ for period $t - 1$, then by $N_{t-1}$ for period $t - 2$, then by $N_{t-2}$ for period $t - 3$ and so on up to the period $t = 0$. This applies to every Consumer population species. By solving Equation (25) in reverse for all the time steps—beginning from the most recent to the initial—the *optimal growth path* is identified with a statistical confidence level as—especially for unstable maps—even the slightest decimal difference will amplify and give different results.

The above apply with the assumption of a *reversible* unstable logistic predator map with knowledge of the optimal growth path upon a record of real-world observations. However, as we discussed in Section 3.2 this is not often the case, as a significant property of unstable maps is their *non-invertibility*, resulting in memory loss [64] if a record of the growth path is unavailable. Unstable logistic cobweb maps are highly sensitive toward the initial conditions (population size $N_0$) [19,23]. As already mentioned, even for unstable maps (Figure 5b), there is *at least one* optimal growth path for every $r > 4$ value and initial population values [15] that can be identified by a Monte Carlo process.

Stabilization of Producer populations at the foundation of the trophic pyramid differentiates from Consumers as they do not rely on any kind of prey availability but on the limiting factor's intensity as described in Equation (2). Essentially, this process has dynamics of its own with solar energy inputs and the engagement of *Decomposers*, although for simplicity, we assume that solar inputs, Decomposers' activity and nutrients' flows are constant. As a result, Equation (25) is modified to adjust to Producers, as Equation (15), to reveal a Producer's optimal growth path. Coefficient *c* must be replaced by coefficient *a* expressing the *limiting factor's consumption intensity* by plant biomass and $N_{n-1_t}$ with $K$ in the formulation of Equation (25), becoming:

$$N_{0_{t-1}} = \frac{K \cdot (1 - r_0 \cdot a_0)}{-r_0 \cdot a_0^2} \tag{26}$$

Equation (26) yields the Producers' optimal stable population for any map (stable or unstable). As Producers are the foundation of the trophic pyramid, their optimal growth path can be found directly by solving Equation (15).

*3.4. Ecological Finance Engineering and Discounting*

In this part, we unify the results of Section 3.1, Section 3.2, Section 3.3 into an integrated NPV model of financial investments in ecosystem services. Our core argument suggests that an ecological finance rationale imposes that the endogenous properties of trophic pyramids' dynamics, such as instability and uncertainty, should be depicted in the NPV model's parameters as *naturally emerging*. As conventional financial instruments usually suffer from many inefficiencies and distortions of the market, the differentiation of ecological finance instruments should at least consist in the adoption of the underlying indices that relate directly to the structure and performance of ecosystems' trophic pyramids, which are the target of the investment, as they unify energetic, ecological and economic aspects [67].

Although our focus is on the part of ecosystem services' *risk-adjusted discounting*, we postulate a generalized NPV model, including the general valuation of ecosystem services, without entering the discussion of how each ecosystem service type should be valued, as this escapes our work's scope. Regarding the *benchmark* or *risk-free* discount rate, a basic debate in the economic literature concerns whether ecological finance instruments should adopt the *market* or the *social discount rate* (SDR) [68,69]. Arguments in favor of the one or the other option have been substantiated from both sides and with a reasonable justification. Indicatively, the supporters of the SDR argue on the grounds that ecosystem services are a *public good*; hence, social time preferences should be taken into account instead of commercial discounting that may suffer from the significant volatility of financial markets and central banks that may wish to prioritize macroeconomic issues without taking into account the thermodynamic foundations of the economic system [57,59]. A counterargument to this rationale focuses on the high demand for private investments; hence, it should be taken into account that private capital operates in terms of *opportunity costs*. If institutional limitations were to be imposed on private capital, then the main goal for its large-scale engagement in ecosystem conservation would be canceled. In addition, the use of private capital for ecosystems' conservation does not signify the privatization of ecosystems *per se* but the return of a fraction of the value of ecosystem services as a payoff for the investor who initially dedicated the funds. Another more elaborate argument is that the scientific knowledge on ecosystems and technology updates constantly in modern societies, and the continuous re-adjustment of the SDR would asymptotically lead it to converge with the market discount rate.

Irrespective of the interesting discussion and the rational arguments by each side, a golden section can be identified in the position that the discount rate should generally be formed without hierarchical interventions. As the most representative of this rationale, we may identify *Hotelling's rule* for benchmark discounting [70]. Specifically, Hotelling's rule concerns the optimal quantity utilization of an *exhaustible* resource (e.g., petroleum) at every time step *t*, taking into account the equilibrium between its market price (*B*), its life-cycle

cost (*C*)—i.e., its extraction, processing, transport, distribution and management at end of life—and the temporal value of money that essentially determines the benchmark discount rate (*i*). According to *Hotelling's rule*, the optimal temporal utilization of an exhaustible resource is achieved when:

$$i = \frac{[dB(t)/dt] - [dC(t)/dt]}{B(t) - C(t)} \tag{27}$$

Equation (27) essentially suggests that the optimal benchmark (risk-free) discount rate achieves a constant temporal equilibrium to the market's weighted percent (%) profit change, as typically, an investor is indifferent between investing its capital in the market or depositing it in a bank. Several economists have improved and extended Hotelling's rule, suggesting that a prerequisite for its optimal function is markets that adopt *total cost accounting* frameworks. Essentially, these are markets that account for *resource depletion* as a cost, as the SEEA and related ecosystem services classification and valuation frameworks suggest [47,48]. Specifically for resource depletion, as the original form of Hotelling's rule addressed exhaustible resources, a solution ought to be found for its application to renewable biological resources, such as forests and fisheries. Indeed, the periodical replenishment of biological stocks is not a sufficient condition for their sustainability as they may be monotonically depleted if the rate of their harvest is higher than the rate of their replenishment. This aspect relates to the *maximum sustainable yield* (MSY) population size depicted in Figure 2 and extends to all kinds of biological stock and ecosystem services. The variable depicting how *net resource depletion* (i.e., when the harvest rate exceeds the replenishment rate) must be incorporated into the total cost is the *scarcity rent* [60] and is usually depicted with the letter ($\lambda$) as a *Lagrange cost multiplier*.

In regard to *risk-adjusted discounting*, an ecosystem's uncertainty is embodied in the discount rate across the long-term programming of ecosystem services' supply. As renewable resources may manifest a quite variable behavior, the discounting should embody the uncertainty of meeting the expected biomass yield and derived revenues. Such cases concern forestry and fishery populations [25,61] as indicative examples of Producers and Consumers. A logistic predator model can include both kinds of populations. As shown in Figures 6 and 7 for *r* = 3.99, intensively oscillating populations embody a high survival risk, which is expressed by the $H(N)_{ADJ}$ index as high uncertainty of the monetary flows that can be produced if they are preserved. Specifically, as the value of ecosystem services derives from the trophic pyramid's biomass, it must be ensured that the risk of biomass collapse and disruption of the flow of ecosystem services is minimized. In short, the conservation of monetary value from ecosystem services' flow must be secured in the future for any kind of population, by compensating capital investors for any increased risk. The *future value* (FV) of an initial investment $K_0$ for a time period *t* is:

$$FV(K_0)_{t:0 \rightarrow T} = K_0 \cdot \left[1 + i \cdot \left(1 + H(N)_{ADJ}\right)\right]^T, \quad i \in R; K, H(N)_{ADJ}, T \in R^+ \tag{28}$$

According to Equation (28), $K_0$ is the initial financial investment at the current time step *t* = 0, *i* is the benchmark or "risk-free" discount rate, and $H(N)_{ADJ}$ is the risk coefficient expressed in terms of Shannon entropy as a positive function of uncertainty caused by the oscillation intensity and unpredictability of the ecosystem's biomass. In Equation (28), we assume $H(X)_{ADJ}$ as expressed in Equation (21) for the normalization of Shannon's entropy to the oscillation range of the target population. In simple words, the *intangible* investment via the use of an ecological financial instrument, such as a "green" bond [8] or a nature-based derivative contract [62] raising capital for building a physical infrastructure to support an ecosystem's conservation, should be compensated for higher risks concerning all of its biomass yields and ecosystem services attached to it. For instance, for a benchmark discount rate *i* = 0.05 (=5%), a financial investment in an ecosystem with chaotic dynamics that is marginally sustainable (e.g., for *r* = 3.99) would require a risk-adjusted discount rate of *i*·(1 + $H(N)_{ADJ}$) = 0.2465 (=24.65%). A same investment on an optimally stable (for

*r* = 2.00) and sustainable population would require only the benchmark discount rate. Typically, the optimally stable population can be considered as the benchmark or *expected* population size, and the adjusted information entropy by Equation (21) can be considered as an additional cost coefficient of deviating from it by the properties of its empirical distribution. Essentially, the adjusted information entropy is an *uncertainty discount cost multiplier*, equal to $H(X)_{ADJ}$ + 1. The physical investment financed by all capital sources has to be profitable by the end of its technical life in terms of its *net present value* (NPV) as:

$$NPV(K)_{t:0 \to T_I \to T_K} = -\sum_{t=0}^{T_I} K_t + \sum_{t=T_I+1}^{T_K} \frac{\sum_{i=1}^{j} \sum_{i=1}^{n} [B_t - (C_t + \lambda_t)]}{\left[1 + i \cdot \left(1 + H(N)_{ADJ}\right)\right]^t}, \quad i \in R; K, H(N)_{ADJ}, T_I, T_K \in R^+ \tag{29}$$

Equation (29) suggests that the *risk-adjusted discounted* profit flows, as the *net sum* of *operational revenues* (*B*) minus the sum of *operational costs* (*C*) and *scarcity rents* (λ) at each time step (*t*) for each ecosystem service *species* (*n*) that belongs to each ecosystem service *class* (*j*) (by the MEA, TEEB or CICES frameworks), for the infrastructure's technical lifetime $T_K$ have to be positive, after compensating for the initial *capital expenses K* across the *preparatory period t:0→$T_I$* before the infrastructure becomes fully operational. The sum of operational costs across the life cycle of each ecosystem service's utilization is augmented by its scarcity rent (expressing *net* resource depletion). The scarcity rent should be accounted for each identified ecosystem service type, as the latter offers a finite amount of tangible or intangible value, irrespective of whether it is a public or private good. For instance, logging may cause a net depletion effect, while as ecosystem service, it could be a private good that can be disposed to anyone who would be willing to pay the declared price, excluding anyone who would not. In contrast, $CO_2$ sequestration tends to function as a public good as it generates a collectively beneficial carrying capacity [71]. However, even if the overall capacity is a collective good, society could decide to optimize its allocation via a market mechanism, where those who deplete more of it would pay more. In any case, and irrespective of the various market mechanisms for allocating the value of ecosystem services in society, our core argument is that even renewable resources are finite and potentially subject to a positive scarcity rent.

Table 2 follows the rationale of Table 1, by presenting the values of the $H(N)_{ADJ}$ index elements for the biomass of *each level* of the trophic pyramid simulated in Appendix A, as well as for the biomass of the whole pyramid, *before* stabilization is applied on the Producer's population. For instance, we see that the adjusted Shannon entropy for the whole pyramid is relatively low due to the very high weight of Producers on total biomass (see Tables A4 and A5), while for specific separate populations, such as secondary to quaternary populations, it is very high. If an ecological finance investment aimed at the conservation of specific trophic levels (compartmentalized investment), we would observe significant variability among the values of the risk-adjusted discount rates. Indicatively, for a benchmark *i* = 5%, the risk-adjusted discount rate for an investment on the Producer would be equal to $i·(1 + H(N)_{ADJ})$ = 6.84%, while for a respective investment on the quaternary Consumer, it would be equal to 12.25%. Although such compartmentalized investments frequently take place for ecosystems with very high diversity (see Section 4.1), the core aspect is the differentiation of risk-adjusted discounting across the increasing uncertainty of population (and biomass) sizes. In contrast, Table 3 presents how the adjusted Shannon entropy is formed *after* the stabilization is applied to the Producer's population, and the uncertainty in the pyramid is significantly reduced.

**Table 2.** Standard and adjusted Shannon entropy values of the observed population sizes at 70 time steps *before* population stabilization at $t = 44$.

| $N_n$ | $r_n \cdot c_n$ | Behavior | (Max − Min)/Max | $H(N)$ | $H(N)_{ADJ}$ |
|-------|-----------------|----------|------------------|--------|--------------|
| $N_0$ | 3.05 | 1-period Oscillating | 0.1965 | 1.8824 | 0.3699 |
| $N_1$ | 2.75 | 1-period Oscillating | 0.4438 | 1.8298 | 0.8121 |
| $N_2$ | 2.15 | 1-period Oscillating | 0.5183 | 2.2176 | 1.1494 |
| $N_3$ | 2.05 | 2/1-period Oscillating | 0.5099 | 2.3658 | 1.2064 |
| $N_4$ | 1.97 | 2/1-period Oscillating | 0.6307 | 2.0906 | 1.3184 |
| $\Sigma N$ | - | 1-period Oscillating | 0.2108 | 2.0512 | 0.4324 |

**Table 3.** Standard and adjusted Shannon entropy values of the observed population sizes at 70 time steps *after* population stabilization at $t = 44$.

| $N_n$ | $r_n \cdot c_n$ | Behavior | (Max − Min)/Max | $H(N)$ | $H(N)_{ADJ}$ |
|-------|-----------------|----------|------------------|--------|--------------|
| $N_0$ | 3.05 | Stable | 0.1550 | 1.6127 | 0.2500 |
| $N_1$ | 2.75 | Convergent Stable | 0.4051 | 1.8991 | 0.7693 |
| $N_2$ | 2.15 | Convergent Stable | 0.4824 | 1.8952 | 0.9142 |
| $N_3$ | 2.05 | Convergent Stable | 0.5257 | 2.2014 | 1.1572 |
| $N_4$ | 1.97 | Convergent Stable | 0.6829 | 2.1235 | 1.4501 |
| $\Sigma N$ | - | Stable | 0.1708 | 1.5736 | 0.2687 |

Table 3 presents the reduced values of the $H(N)_{ADJ}$ index elements as a result of the Producers' population stabilization, based on Equation (26) and analytically described in Appendix A. In terms of $H(N)_{ADJ}$ values the percent reductions range from ~4% for tertiary Consumers (lowest) to 37.85% for the total biomass in the pyramid (on which Producers with a 32.41% reduction have a very high weight; see Tables A4 and A5, respectively). The only exception to the above view is quaternary Consumers with a ~10% increase. By a first look this signifies an increase in the population's size uncertainty, which would be contradictory to the rationale of population stabilization. In addition, as it can be seen from Figure A10, the quaternary Consumer's population size increases after the stabilization; however, as the number of the simulation's time steps is quite low, this increase adds to its variability, although the overall effect is positive (biomass increase). If the number of time steps asymptotically approached infinity, the quaternary Consumer's population would manifest similar behavior to all other populations in the pyramid.

Another critical aspect regarding the NPV is that higher $H(N)_{ADJ}$ coefficients signify that ecosystem services' profits are highly uncertain or that at least one risk-adjusted alternative (as opportunity cost) investment has higher value. In short, the monetary value of ecosystem services discounted in the future is considered lower when conserved into the ecosystem's biomass. This condition under the constant pressure of global biodiversity depletion could result to a severe deficit of financial capital on ecosystem services' conservation. In addition, authors have argued that increased survival risk could lead to intense competition on biomass over-harvesting in order to avoid taking over the potential future risk of ecosystem biomass collapses and the consequent loss of value [61]. In such extreme game theory cases that are frequently combined to the phenomenon of the *Tragedy of Commons* [72,73], it is preferred that ecosystem biomass is over-harvested in the present so that present revenues are stored in alternative investments with higher future yield. Hence, besides high discount rates as compensation for high-risk investments, institutional constraints and penalties on invested capital should accompany such financial schemes to prevent moral hazard behaviors and sustainably conserve the value generated by trophic pyramids. Additionally, physical investments on ecosystem services may include population stabilization. Eventually, the combination of population control and the engineering of ecological finance instruments aims at the minimum deviation (or in some cases even the convergence) of the benchmark and risk-adjusted discount rates for the reduction in risk-related costs of investments in ecosystem services.

## 4. Discussion

In the Discussion, we examine two critical pillars of our work's possible extensions on the ecological energetics of trophic pyramids, their logistic predation dynamic modeling and human-induced risks on ecosystem services. In particular, we briefly discuss the special issue of *diversity* in a trophic pyramid, its energetic limits and its relation to specific *competition types* suggested by ecologists. In turn, we upscale this discussion by including the aspect of *human environmental migration* in the context of the trophic pyramid concept and its potential risks for ecosystem services' conservation.

### 4.1. Energy, Diversity and Competition

A significant application of the concept of information entropy in ecosystems concerns *biodiversity*, as it comprises a vital pillar for ecosystem management. An emerging issue concerns the energetic constraints of trophic pyramids, by Lindeman's rule, on the diversity indices used in ecology. A fundamental and straightforward diversity index is the *Herfindahl–Hirschman* index [74]:

$$D_{H-H} = \sum_{i=1}^{n} \left( \frac{N_i}{\sum\limits_{i=1}^{n} N_i} \right)^2, \quad N_i \in [0, +\infty), D_{H-H} \in [0,1] \tag{30}$$

Equation (30) essentially yields the squared sum of the shares of populations of the various species in the ecosystem to the total ecosystem population. In other versions, the index is used with a square root, which tends to yield higher values. The lower the $D_{H-H}$ value, the higher the biodiversity, and vice versa. Alternatively, the *Simpson* index [75] is widely used in ecology:

$$D_S = \frac{\sum\limits_{i=1}^{n} [N_i \cdot (N_i - 1)]}{\sum\limits_{i=1}^{n} N_i \cdot \left[ \left( \sum\limits_{i=1}^{n} N_i \right) - 1 \right]}, \quad N_i \in [0, +\infty), D_S \in [0,1] \tag{31}$$

Similarly to the $D_{H-H}$, the lower the value of $D_S$, the higher the biodiversity. For very large populations, both indices asymptotically converge, giving the same result; hence, they are often used interchangeably. Another common feature is that both indices take values in the range [0,1]. Except for measuring ecosystems' biodiversity, both indices are widely used in other fields, such as economics, for estimating markets' concentration and the respective level of industrial competition or its tendency toward oligopoly or monopoly. Indicatively, the $D_{H-H}$ along with a set of derivative indices has been developed to measure production cost diversity in *industrial ecosystems* [58].

In Table 4, we present the application of the $D_{H-H}$ to global biomass fractions [34] and to stabilized populations of the simulation in Appendix A. The $D_{H-H}$ values of Table 4 indicate low *vertical* biodiversity, meaning that across the global taxonomy (for global data) and the trophic pyramid (for the simulation), the total biomass is unequally concentrated between taxonomies and trophic levels, although the intra-taxonomy application of the $D_{H-H}$ could suggest high diversity. The core aspect here is that the energy efficiency constraints of the Eltonian Pyramid pose a respective constraint on the potential vertical diversity. Specifically, as biomass is essentially a secondary storable form of incoming solar energy [15], according to Lindeman's rule, for a three-level trophic pyramid (Producer, Herbivore, Carnivore), the required energy for sustaining one top predator individual in terms of solar energy (by Equation (7)) is of an order of $10^4$. Additionally, scholars suggest [76] that phytomass has significant power generation variability—affected by plant species' diversity—that ranges from 0.1 to 10 W/m$^2$ for a respective range of land use between 1m$^2$ and 1km$^2$, nesting phytomass power density in the range of $10^{-4}$ W/m$^2$–1 W/m$^2$.

With just a 10% transformation efficiency of solar energy by Producers, the energy limits of the pyramid's upper levels are set from its foundations.

**Table 4.** $D_{H-H}$ index values for global biodiversity ($G$) as presented in Figure 4 (the 10 classes from Arthropods to Wild Birds belong to *Animal Biomass*) and the simulated trophic pyramid ($S$).

| $N_n/\sum N_G$ | $(N_n/\sum N_G)^2$ | $N_n/\sum N_S$ | $(N_n/\sum N_S)^2$ | $D_{H-H(G)}$ | $D_{H-H(S)}$ |
|---|---|---|---|---|---|
| Plants | 0.67898 | Producer | 0.87799 | - | - |
| Bacteria | 0.01638 | 1st Consumer | 0.00356 | - | - |
| Fungi | 0.00048 | 2nd Consumer | 0.00001 | - | - |
| Archea | 0.00023 | 3rd Consumer | $2.6 \times 10^{-8}$ | - | - |
| Protists | 0.00005 | 4th Consumer | $6.4 \times 10^{-11}$ | - | - |
| Animals | 0.00002 | - | - | - | - |
| $\sum(N_n/\sum N)^2$ | - | - | - | **0.6961** | **0.8816** |

Moreover, as Producers are the only level able to transform solar energy directly to biomass (while other levels have to prey on at least one level below them), they essentially comprise a pivotal level and solar energy transformation hub. Hence, trophic topologies between Producers and all upper pyramid levels are expressed by either *linear* (when an upper level preys on the exact lower) or *mesh* (when upper levels prey on more than one lower level) relations. This establishes Producers as the unique intermediates of primary solar energy inflow and a common energy currency for all upper pyramid levels. Based on the above, Producers become even more essential to the trophic pyramid if we take into consideration the intensity of limiting factors, by Equation (2), that determine fundamentally both the total energy quantity and its flow per unit time. As nutrients and abiotic factors (e.g., sunlight, water, physical space) are combined with high *complementarity* [15], extreme scarcity in one nutrient cannot be compensated by over-availability in another. Thus, an intense limiting factor on Producers would limit the population sizes in all upper pyramid levels.

Another critical aspect of the Eltonian Pyramid's energetic constraints concerns the two possible versions of predation patterns, distinguishing *contest* from *scramble* competition. Essentially, the above structure suggests that any energy flow in the pyramid is a "zero-sum game" as the gain of one level is the loss of another. For the case of the *linear topology* (only $n \rightarrow n-1$ predation), energy flows are compartmentalized, and the population growth limitations on each trophic level are more intensive under the lack of prey alternatives. For such predation topologies, we would expect that the trophic pyramid and the logistic predator model tend to manifest a *contest competition* behavior. In contest competition patterns, population stabilization is achieved endogenously via individuals competing for the exclusivity of available resources. Hence, the "winners" would have access to all of the required energy to form biomass and reproduce. In contrast, the "losers" would receive nothing and perish. With this structure, contest competition manifests higher stability that in its optimal mathematical form is expressed by the Beverton–Holt model [15,28,30]. In contrast, for *mesh topologies* ($n \rightarrow m \forall m < n$ predation), it is highly possible that the system manifests elements of *scramble competition* expressed by the Ricker model [15,29]. Such a pattern suggests that biomass is distributed more equally among predators preying on all lower pyramid levels, however, frequently at the cost of gaining lower energy than the minimum required for their maintenance and reproduction (partially this could be further explained as the low energy return on the energy expenditure for hunting when the prey is of small size to cover the predator's needs). However, the Ricker map is highly resilient even for very large $r$ parameter values that lead to intense oscillations between very small and very large population sizes [15]. As argued [15], the logistic cobweb map (as the basis for the logistic predator map) combines properties from both the Ricker and the Beverton–Holt models. While oscillations for $r > 2$ at any pyramid level would signify the existence of scramble competition, for $1 < r < 2$, they manifest stable behavior of

Beverton–Holt type (where any oscillations observed in such cases occur due to $r > 2$ value in at least one lower pyramid level that comprises prey/carrying capacity).

### 4.2. Humans in the Eltonian Pyramid, Environmental Migration and Competition

Endogenously oscillating trophic pyramids are highly sensitive against exogenous amplifications. This sensitivity has numerous extensions beyond the cost of natural capital conservation (as discussed in Section 3.4), extending to human environmental migration across the collapse of local ecosystems and their services. The collapse of a local ecosystem would mean that human populations will be deprived of ecosystem services that are essential for their survival, be forced to relocate and put additional environmental pressure on lands in which they arrive [77], especially if these lands are already in amarginal viability state. Indeed, considering the already high global HANPP average estimations [51], it is easily understood that the distribution of HANPP pressures across the world is highly unequal. Some areas with rich biomass, biodiversity and established institutions on sustainable ecosystem management may manifest very low HANPP values, while other areas may manifest a near-depletion state, which intensifies in cases of natural disasters, such as wildfires [78] that reduce biomass primary productivity (also as constituent of the HANPP index). In addition, stressors on biomass primary productivity have indirect impacts on environmental politics' pillars, such as the *European Green Deal* (EGD) [79]. For instance, *farm to fork* (F2F) strategies, as a pivotal element of the EGD, aim at the EU's agriculture structural shift toward regenerative practices [80] that highly depend on biodiversity and ecosystem services, while reducing the life-cycle HANPP. Additional stressors on trophic pyramids and biomass productivities would partially negate this effect, especially in Mediterranean countries [79] suffering from both high wildfire risks [78] and flood risks. Such a case is Greece, where its agricultural infrastructure was severely impacted by the summer 2023 wildfires and the subsequent floods. However, these effects expand even in less marginal ecosystem states. Indicatively, for the energy sector, societies may deal with high scarcity rents on ecosystem services due to inefficient energy transitions toward *renewable energy sources*(RES) [81,82] that would limit $CO_2$ carrying capacity consumption and depletion rates.

Human populations essentially comprise a part of the trophic pyramid, although the practice of predation may not be the dominant pattern of securing the minimum energy flow requirements for their societies. Even across the transition from hunter-gatherer to agrarian societies with large-scale domestication of animals, the trophic pyramid concept was preserved [15]. In any case, within the Eltonian Pyramid context, humans are positioned as the top predator, depending on the constant flow of ecosystem services that are generated in the lower pyramid levels. Scientific-based ecosystem management aims at securing the long-term flow of ecosystem services that are vital for the long-term welfare of human societies [63]. Benefits from population control and ecosystem stabilization are consistent with this target. Via an anthropological approach, we may suggest that human population concentrations are shaped historically as a function of biomass and availability of services by the ecosystem in which they decide to settle [35,36]. Human settlements historically had the top consumer role in the Eltonian Pyramid, depending on the availability of biomass and ecosystem services produced by the lower levels that formed favorable health conditions for settling, such as air and water purification, favorable micro-climate regulation and constant food flow. In this case, the rest of the species of the Eltonian Pyramid constituted the carrying capacity of human populations. A stable carrying capacity in the long term would allow reproduction programming, while intensive and frequent biomass oscillations would probably lead a part of the human population to reduction or/and environmental migration.

Although environmental migration may mitigate survival risks locally, it eliminates them globally only under specific conditions. These conditions concern the consumption of residual carrying capacity (e.g., in HANPP terms) of the areas where migrating populations relocate. Historically, environmental migration events have usually constituted a factor of

instability to ecosystems receiving migrating populations as they had to endure further pressure [76]. If the carrying capacity of the ecosystem that receives the migrating populations is insufficient to sustain their numbers, resource conflicts are very likely to occur between settled and settling populations. In ecological modelling terms, this could result in either a new environmental overshoot deriving from *scramble* competition or a *contest* competition process, manifested as a war that would restore the ecological equilibrium with one winner side and lower human population sizes. Periodical environmental migration patterns, such as in South-East Asia due to the monsoons, can be considered endogenous and predictable. Such patterns are controllable and distinguished by relatively stable successions of population movements and carrying capacities' congestion and de-congestion cycles at small spatial scales. In contrast, environmental migration of large spatio-temporal scales (regional, continental or global) comprises a potentially permanent and irreversible ecological stressor and risk factor. For such cases, ecosystem management becomes of primary global concern as the size of environmentally displaced people is expected to increase by 2050 [76].

## 5. Conclusions

Our work develops an argumentation that the capacity of an ecosystem to provide valuable services with high economic value for humanity fundamentally depends on the stability of populations of *trophic pyramid* levels. We begin in *Materials and Methods* with a review of logistic growth functions for both continuous and discrete time and substantiate our preference for the latter. We then argue on the emergence of the *carrying capacity* from the *limiting factor*. We continue with an analysis of the *Eltonian Pyramid* notion accompanied by global data on biomass shares of the various taxonomies to total planetary biomass. Global data suggest that plant biomass dominates the biosphere. In this context, we present the conveniences of the *logistic cobweb map* on which we develop our *logistic predator* model. Across the model's formulation, we focused on its biophysical consistency with the Eltonian Pyramid concept and Lindeman's rule of 10% metabolic efficiency that puts an upper limit on biomass accumulation at each level of the trophic pyramid. Thus, we manage to show that logistic predator models embody higher flexibility than conventional prey–predator models, as they incorporate features from the logistic map and are able to depict a higher variety of prey–predator dynamics. As our core assumption is that biomass in trophic pyramids comprises the source of ecosystem services' production, we close this section by presenting the background of the MEA, TEEB and CICES classification and valuation frameworks of ecosystem services and their relation to the ongoing establishment of "green" national and corporate accounts, along with empirical data on the correlation of global biomass primary production to ecosystem services' value.

In our *Results* section, we present the mathematical conditions for the *stability* and *sustainability* of the logistic cobweb map as the foundation of the logistic predator model. This part is directly related to our core work hypothesis that *the uncertainty of biomass in trophic pyramids has to be incorporated in the risk-adjusted discounting of ecological finance instruments*. We show that the critical parameter for the stability of populations in trophic pyramids is the intrinsic growth ratio. On these grounds, we discuss thoroughly in the second part of this section the relation between *information entropy* as a measure of statistical uncertainty and the *value of natural capital*. Specifically, we develop a statistical context, showing that statistical uncertainty has a negative effect on natural capital supply and value, as it contains higher sustainability risks. This uncertainty is *naturally emerging* from the endogenous instability of trophic pyramids. To depict this effect, we postulate an *adjusted Shannon entropy* index $H(N)_{ADJ}$ that comprises the risk coefficient added to the benchmark discount rate and is suitable for trophic pyramid applications. We continue with the mathematical conditions for the stabilization of oscillating pyramids via population control. As we demonstrate, when the value of the intrinsic growth rate parameter $r$ is excessively high, giving unstable or unsustainable populations, spontaneous stabilization is *highly improbable*, requiring external population control to reduce uncertainty. To substantiate

our findings, we perform a Monte Carlo numerical simulation of a logistic predator model with a five-level trophic pyramid, where we apply population stabilization and measure its effect on the $H(N)_{ADJ}$. Finally, in the concluding part of our results, we integrate the above elements in a *net present value* (NPV) model of ecosystem services valuation and discounting. Here, we present the effect of population control on the reduction in the $H(N)_{ADJ}$ (risk) coefficient and its long-term economic value via lower discount rates on financial capital invested in ecosystems' conservation.

In our *Discussion*, we examine two extensions of our work. We first discuss the effect that Lindeman's energetic constraints have on the Eltonian Pyramid's *vertical diversity*. For this purpose, we use the *Herfindahl–Hirschman* index to measure global taxonomical diversity [34] and elaborate on the inevitable dominance of plant biomass as a hub of direct solar energy transformation. On these grounds, we further discuss issues of competition typology of Consumer populations in trophic pyramids. The second pillar of this section concerns human populations as the top level of the Eltonian Pyramid and their historical dependence on ecosystem goods and services produced by the lower levels, irrespective of the fact that this concept was not realized. In an ecological modeling context, we then discuss the critical aspect of *environmental migration* as a long-term persistent ecological stressor and risk factor.

Finally, continuing this foundational work, we identify our next major research and modeling challenges. Due to its already large size, our work leaves several research and mathematical modeling aspects for a future paper. Such aspects concern the following: (a) The postulation of parsimonious *stochastic* versions of the logistic predator model to include the background uncertainty of abiotic factors (e.g., solar energy, nutrients' flow) that determine fundamentally Producers' growth in the trophic pyramid. In our current version, those factors were considered constant for simplicity; however, in real ecosystems, they are variable and may establish a *minimum* $H(N)_{ADJ}$ value. (b) The formulation of logistic predator *typologies* according to *mutual information* metrics. For trophic pyramid levels with high statistical independence, the cost of stabilization is expected to be higher due to the need to apply population control at more than one level. (c) The generalized reformulation of the $H(N)_{ADJ}$ index to include the weighted average of each pyramid level's biomass contribution to total biomass for compartmentalized investments, as a special case briefly discussed in Section 3.4. (d) The inclusion of other information entropy metrics in the valuation and discounting of ecosystem services by trophic pyramid typology (e.g., with high or low diversity). (e) In addition to challenge (a), the generalization of the logistic predator model to include the participation of *Decomposers* in a closed-loop circular model. (f) Extending the logistic predator modeling to *mesh* predation topologies within a stochastic framework and examining their effect on the size and stability of the trophic pyramid, as well as on the discount rate of ecosystem conservation investments. (g) The *price volatility* estimation of ecosystem services across the population fluctuations of the trophic pyramid.

**Author Contributions:** Conceptualization, analytical framework, modeling, methodology, G.K.; formal analysis and writing, G.K. and N.M.; visualization, G.K.; manuscript review and refinement, G.K. and N.M.; research, G.K.; manuscript supervision and quality control, N.M. All authors have read and agreed to the published version of the manuscript.

**Funding:** The research was conducted exclusively by the authors with their own time resources. The authors received no external funding for the research and the manuscript writing.

**Data Availability Statement:** All raw data used for this work are publicly available and are referenced in the "References" section. The provision of processed data will be examined by the authors upon request.

**Acknowledgments:** The help and support of the MDPI Land Editorial Office as well as of the Special Issue's Guest Editors is highly appreciated.

**Conflicts of Interest:** The authors declare no conflict of interest.

## Appendix A. Simulation Analysis

We develop a simple numerical example of how logistic predator dynamics evolve by the intrinsic growth rate parameter values and the impact of stabilization from targeted population control for an oscillating trophic pyramid, via a Monte Carlo simulation for five levels (levels $0\rightarrow4$) with a *linear* energy flow topology, meaning that any trophic level $n > 0$ seeks prey *exclusively* in the exact lower level $n - 1$. Hence, the simulation is based mainly on Equation (11) for *Consumer* levels ($n > 1$) and Equation (15) for *Producers* ($n = 0$), constrained by Equations (12) and (13) for Consumers and by Equation (2) for Producers.

*Appendix A.1. Methodology of the Simulation Process*

All intermediate populations (levels $1\rightarrow3$) interact by two, meaning that each one of them is *both a predator and a prey*. The $n = 0$ depicts Producers as prey exclusively for the $n = 1$ (herbivore), while the $n = 4$ (top predator) only preys upon the $n = 3$. For each $n$-level population, *carrying capacity* is variable and equal to the availability of the $n - 1$ level population, except for Producers, which is equal to $K$ and assumed constant. The limiting factor is presented by $c_0 = a_0 = 2$. Each population begins with an initial positive size $N_{n0}$. The parameters for all species are presented in Table A1 below:

**Table A1.** The logistic predator model's parameters $r$, $c$, $K$ and the starting population value $N_{n0}$ for each $n$-level population with $n = 4$.

| Model Parameters | | | | | |
|---|---|---|---|---|---|
| $n$ | $r_n$ | $c_n$ | $r_n \cdot c_n$ | $N_{n0}$ | $K$ |
| 0 | 1.525 | 2 | 3.05 | 1 | 1,000,000 |
| 1 | 0.275 | 10 | 2.75 | >0 | --- |
| 2 | 0.215 | 10 | 2.15 | >0 | --- |
| 3 | 0.205 | 10 | 2.05 | >0 | --- |
| 4 | 0.197 | 10 | 1.97 | >0 | --- |

In Table A1, the net intrinsic growth rates $r$ are reduced to the intrinsic growth rate of the *highest reproducing species*, as developed in Section 2.4 and Equation (11). Population sizes for every species at each level of the trophic pyramid for 70 time steps are presented in Table A2. For simplicity, we assume that population units $N$ are equal to biomass units. All units presented in the tables below can be considered to be of any scale ($10^2$, $10^3$, $10^6$, etc.). According to Equations (11), (13) and (15), initial sizes of populations $N_{n0}$ are repeated until they yield non-zero sizes (values in borders). $N = 0$ population sizes for any $n$-level mean that the availability of the $n - 1$ level population is insufficient to sustain the $n$ level population. Explanations for the natural meaning of this phenomenon refer to the lack of biodiversity at the $n - 1$ level for an ecosystem under formation [12,36,42]. Consumer species are not observed until Producers' biomass is sufficient to support a minimum population; consequently, this forbids the arrival of predators in the upper pyramid levels as well. Hence, the trophic pyramid is built step by step, initially populated only with primary consumers. As the trophic pyramid reaches its maturity, species of bigger biomass and energy needs appear (presumably migrating from nearby ecosystems). As the $n - 1$ population grows, the $n$ population becomes positive and also grows upon the availability of prey (carrying capacity). Once that happens, the first non-zero value replaces the initial value $N_n = 0$ in the model in the next time step, for the extraction of the next population (e.g., the value $N_1 = 1.62$ in $t = 3$ replaces the value $N_1 = 0$ in $t = 2$, according to Equations (11) and (13), in order to yield the population $N_1 = 3.63$ in $t = 4$). Also, population ratios must comply to the constraints of Equation (13) for consistence withthe notion of the Eltonian Pyramid. By the above rules, the simulated formation and growth dynamics of the whole trophic pyramid are presented in Table A2 below:

**Table A2.** Simulation results of the logistic predator model for 70 time steps for each *n*-level population as well as for the total trophic pyramid biomass ($\sum N_{nt}$).

| | | | Logistic Predator Dynamics (Before $N_0$ Stabilization) | | | |
|---|---|---|---|---|---|---|
| *t* | $N_{0t}$ | $N_{1t}$ | $N_{2t}$ | $N_{3t}$ | $N_{4t}$ | $\sum(N_{nt})$ |
| 1 | 3.05 | 0.00 | 0.00 | 0.00 | 0.00 | 3.05 |
| 2 | 9.30 | 0.00 | 0.00 | 0.00 | 0.00 | 9.30 |
| 3 | 28.37 | 1.62 | 0.00 | 0.00 | 0.00 | 29.99 |
| 4 | 86.53 | 3.63 | 0.00 | 0.00 | 0.00 | 90.16 |
| 5 | 263.87 | 8.60 | 0.00 | 0.00 | 0.00 | 272.47 |
| 6 | 804.37 | 21.12 | 0.29 | 0.00 | 0.00 | 825.79 |
| 7 | 2449.40 | 53.08 | 3.43 | 0.00 | 0.00 | 2505.90 |
| 8 | 7434.06 | 135.55 | 5.51 | 0.00 | 0.00 | 7575.12 |
| 9 | 22,336.77 | 350.14 | 9.98 | 0.00 | 0.00 | 22,696.88 |
| 10 | 65,083.66 | 911.09 | 19.10 | 0.00 | 0.00 | 66,013.84 |
| 11 | 172,666.27 | 2373.29 | 37.76 | 1.93 | 0.00 | 175,079.24 |
| 12 | 344,768.91 | 6077.27 | 76.14 | 2.95 | 0.00 | 350,925.27 |
| 13 | 326,465.00 | 13,601.40 | 154.54 | 4.90 | 0.00 | 340,225.84 |
| 14 | 345,583.93 | 22,682.54 | 309.62 | 8.45 | 0.00 | 368,584.54 |
| 15 | 325,518.65 | 18,911.93 | 556.71 | 14.69 | 0.00 | 345,001.98 |
| 16 | 346,461.30 | 23,618.83 | 914.80 | 25.28 | 0.82 | 371,021.04 |
| 17 | 324,490.83 | 17,675.09 | 948.86 | 38.02 | 1.27 | 343,154.07 |
| 18 | 347,401.81 | 23,876.51 | 1229.33 | 53.83 | 1.91 | 372,563.39 |
| 19 | 323,378.62 | 17,180.35 | 751.83 | 31.34 | 1.47 | 341,343.60 |
| 20 | 348,405.03 | 23,948.30 | 1108.98 | 46.09 | 1.97 | 373,510.37 |
| 21 | 322,180.34 | 16,904.40 | 820.13 | 41.39 | 2.03 | 339,948.30 |
| 22 | 349,468.99 | 24,000.51 | 1160.74 | 54.59 | 2.51 | 374,687.35 |
| 23 | 320,896.12 | 16,637.53 | 754.50 | 30.94 | 0.93 | 338,320.02 |
| 24 | 350,589.82 | 24,040.64 | 1113.07 | 45.80 | 1.46 | 375,790.79 |
| 25 | 319,528.30 | 16,370.65 | 765.99 | 37.75 | 1.76 | 336,704.45 |
| 26 | 351,761.48 | 24,067.73 | 1122.73 | 51.37 | 2.28 | 377,005.59 |
| 27 | 318,082.07 | 16,106.32 | 731.22 | 31.33 | 1.22 | 334,952.16 |
| 28 | 352,975.47 | 24,081.70 | 1094.76 | 45.84 | 1.77 | 378,199.55 |
| 29 | 316,565.92 | 15,846.45 | 727.64 | 34.77 | 1.71 | 333,176.49 |
| 30 | 354,220.77 | 24,082.76 | 1091.75 | 48.58 | 2.18 | 379,446.05 |
| 31 | 314,991.99 | 15,593.18 | 703.84 | 30.85 | 1.26 | 331,321.12 |
| 32 | 355,483.85 | 24,071.53 | 1070.78 | 45.02 | 1.78 | 380,672.97 |
| 33 | 313,376.26 | 15,348.69 | 696.09 | 32.60 | 1.59 | 329,455.24 |
| 34 | 356,749.04 | 24,049.04 | 1063.41 | 46.34 | 2.06 | 381,909.89 |
| 35 | 311,738.32 | 15,115.15 | 677.81 | 30.05 | 1.28 | 327,562.60 |
| 36 | 357,999.12 | 24,016.72 | 1046.00 | 43.90 | 1.78 | 383,107.53 |
| 37 | 310,100.76 | 14,894.61 | 669.57 | 30.99 | 1.49 | 325,697.41 |
| 38 | 359,216.18 | 23,976.37 | 1037.55 | 44.55 | 1.95 | 384,276.61 |
| 39 | 308,488.13 | 14,688.89 | 655.05 | 29.21 | 1.27 | 323,862.56 |
| 40 | 360,382.74 | 23,930.02 | 1022.84 | 42.78 | 1.76 | 385,380.15 |
| 41 | 306,925.47 | 14,499.46 | 647.78 | 29.78 | 1.42 | 322,103.91 |
| 42 | 361,482.90 | 23,879.83 | 1014.93 | 43.14 | 1.87 | 386,422.67 |
| 43 | 305,436.54 | 14,327.37 | 636.33 | 28.48 | 1.26 | 320,429.99 |
| 44 | 362,503.42 | 23,827.94 | 1002.75 | 41.81 | 1.74 | 387,377.66 |
| 45 | 304,042.18 | 14,173.12 | 630.60 | 28.89 | 1.36 | 318,876.15 |
| 46 | 363,434.60 | 23,776.30 | 996.20 | 42.05 | 1.82 | 388,250.96 |
| 47 | 302,758.81 | 14,036.72 | 621.75 | 27.91 | 1.25 | 317,446.43 |
| 48 | 364,270.70 | 23,726.57 | 986.47 | 41.02 | 1.71 | 389,026.47 |
| 49 | 301,597.46 | 13,917.64 | 617.63 | 28.24 | 1.33 | 316,162.30 |
| 50 | 365,009.98 | 23,680.02 | 981.56 | 41.24 | 1.77 | 389,714.57 |
| 51 | 300,563.50 | 13,814.95 | 610.94 | 27.48 | 1.24 | 315,018.10 |
| 52 | 365,654.33 | 23,637.52 | 974.02 | 40.44 | 1.69 | 390,308.01 |
| 53 | 299,656.86 | 13,727.40 | 608.25 | 27.79 | 1.30 | 314,021.60 |
| 54 | 366,208.60 | 23,599.56 | 970.69 | 40.66 | 1.74 | 390,821.24 |
| 55 | 298,872.93 | 13,653.51 | 603.26 | 27.18 | 1.23 | 313,158.10 |

**Table A2.** *Cont.*

| | Logistic Predator Dynamics (Before $N_0$ Stabilization) | | | | | |
|---|---|---|---|---|---|---|
| $t$ | $N_{0t}$ | $N_{1t}$ | $N_{2t}$ | $N_{3t}$ | $N_{4t}$ | $\sum(N_{nt})$ |
| 56 | 366,679.76 | 23,566.28 | 964.99 | 40.02 | 1.68 | 391,252.73 |
| 57 | 298,203.58 | 13,591.72 | 601.70 | 27.47 | 1.29 | 312,425.76 |
| 58 | 367,076.13 | 23,537.57 | 962.96 | 40.25 | 1.72 | 391,618.64 |
| 59 | 297,638.40 | 13,540.46 | 597.98 | 26.97 | 1.23 | 311,805.03 |
| 60 | 367,406.56 | 23,513.15 | 958.69 | 39.74 | 1.67 | 391,919.81 |
| 61 | 297,165.77 | 13,498.22 | 597.26 | 27.26 | 1.27 | 311,289.79 |
| 62 | 367,679.88 | 23,492.61 | 957.64 | 39.98 | 1.71 | 392,171.83 |
| 63 | 296,773.82 | 13,463.62 | 594.45 | 26.84 | 1.22 | 310,859.95 |
| 64 | 367,904.48 | 23,475.52 | 954.43 | 39.55 | 1.66 | 392,375.64 |
| 65 | 296,451.06 | 13,435.42 | 594.30 | 27.12 | 1.27 | 310,509.16 |
| 66 | 368,088.03 | 23,461.39 | 954.08 | 39.80 | 1.70 | 392,544.99 |
| 67 | 296,186.83 | 13,412.52 | 592.13 | 26.75 | 1.22 | 310,219.46 |
| 68 | 368,237.34 | 23,449.80 | 951.62 | 39.43 | 1.66 | 392,679.85 |
| 69 | 295,971.58 | 13,394.00 | 592.35 | 27.03 | 1.26 | 309,986.22 |
| 70 | 368,358.34 | 23,440.35 | 951.72 | 39.67 | 1.70 | 392,791.77 |

Respectively to the boxed values in Table A2 for the first positive population sizes, time step $t^*$ signifies with grey font the first population size after which oscillations begin $\forall t > t^*$. For each level $n$ until $t^*$, the population path is monotonically growing; hence, the part we are interested in for the stabilization via population control is the oscillating part for $t:t^* \rightarrow 70$. For levels $n > 0$, oscillations occur by two determinants: (a) an *exogenous* variable carrying capacity $\forall n > 0$ and (b) an *endogenous* intrinsic growth rate parameter values $r_n \cdot c_n$, as presented in Table A1 and following the rules of Section 2.4. The growth dynamics of each $n$-level population *before* stabilization are presented in Figures A1–A5 below:

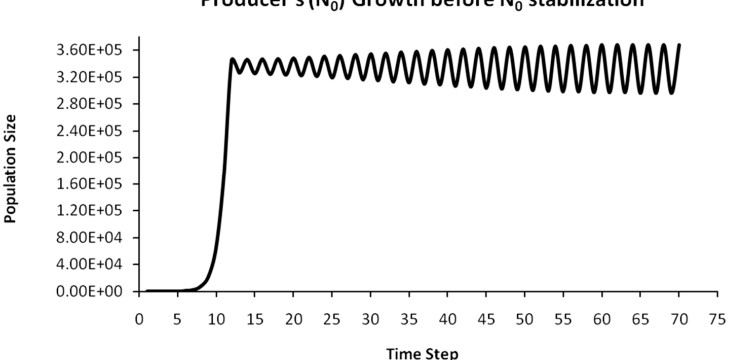

**Figure A1.** Growth dynamics of population $N_0$ (Producer).

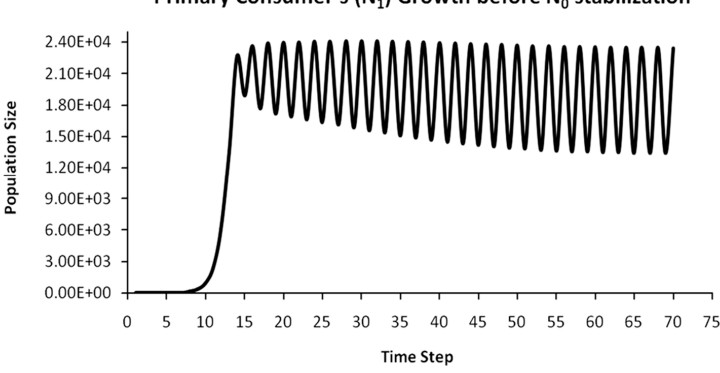

**Figure A2.** Growth dynamics of population $N_1$ (Primary Consumer or Herbivore).

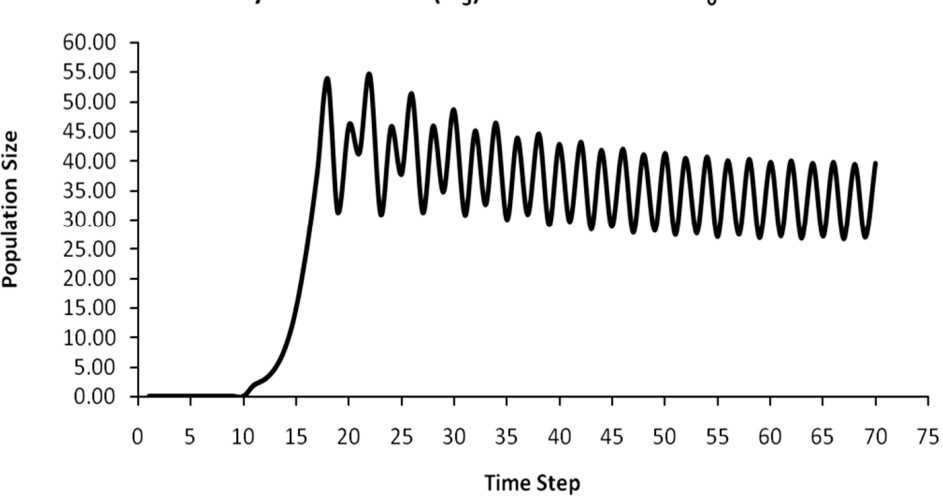

**Figure A3.** Growth dynamics of population $N_2$ (Secondary Consumer or Lowest Predator).

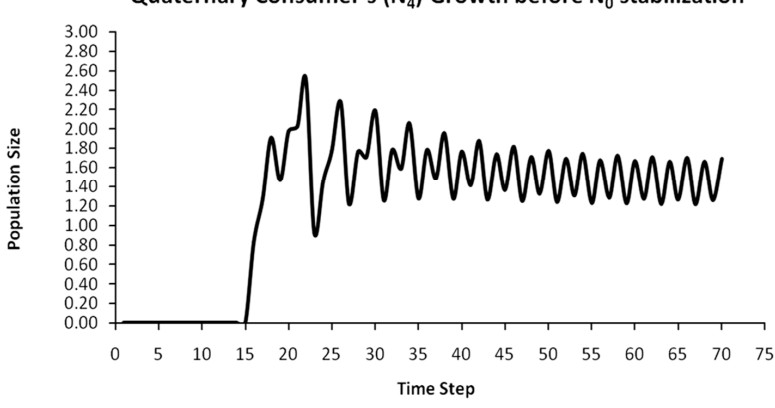

**Figure A4.** Growth dynamics of population $N_3$ (Tertiary Consumer or Intermediate Predator).

Quaternary Consumer's ($N_4$) Growth before $N_0$ stabilization

**Figure A5.** Growth dynamics of population $N_4$ (Quaternary Consumer or Top Predator).

*Appendix A.2. Stabilizing the Trophic Pyramid via Control of the Key Population*

Each trophic pyramid level *n* level oscilates both exogenously and endogenously: in the first case, due to variable carrying capacities $\forall n > 0$ and in the second, due to intrinsic

growth rate parameter values $r_n \cdot c_n > 2 \ \forall n \in [0,3]$. In biophysical terms, this means that even after the Producer's stabilizationas as *key population* (transferring oscillations to all upper pyramid levels) that would create a constant carrying capacity for the $n_1$ level, some endogenous residual oscillations remain for all levels except for the top predator. From that point and after, as described in Table 3, endogenous oscillations will further depend on the deviation of the intrinsic growth rate from the optimal one ($r_n \cdot c_n = 2$), either bing convergent stable ($2 < r_n \cdot c_n \leq 3$), oscillating in stable periods ($3 < r_n \cdot c_n \leq 3.5$) or chaotic and marginally sustainable ($3.5 < r_n \cdot c_n \leq 3.99$).

Producers are the cause for the transfer of oscillations to all upper levels $n > 0$ as the first pyramid level with endogenous instability. With the assumption of linear topology of energy flows, to eliminate exogenous oscillations to all $n > 0$ levels, population control is required at the $n = 0$ level, establishing Producers as a *key population*, with its control assumed to occur at time step $t^* = 44$, separating the period from the first oscillation $t^*$ for the population with the biggest weight in total biomass (which is $t = 12$ for the Producer or $N_0$) in two equal parts up to the last time step ($t = 70$). Normally, the population size of $N_0$ at period $t = 44$ is by Table A2 equal to 362,503.42 individuals; however, by reducing this size by 24,437.85 individuals, the population size becomes 336,065.57 units (see Table A3). By feeding this value to the next time step, the $N_0$ will remain *long-term stable* (although not necessarily perpetually) transferring partial stability to all $n > 0$ levels. Table A3 presents the updated population dynamics for $t:40 \rightarrow 70 \forall n \in [0,4]$:

**Table A3.** Stabilization of population sizes for all $n$-levels after applying targeted population control on the Producer as *key population* at time step $t = 44$.

| | Logistic Predator Dynamics (After $N_0$ Stabilization). | | | | | |
|---|---|---|---|---|---|---|
| $t$ | $N_{0t}$ | $N_{1t}$ | $N_{2t}$ | $N_{3t}$ | $N_{4t}$ | $\sum(N_{nt})$ |
| 40 | 360,382.74 | 23,930.02 | 1022.84 | 42.78 | 1.76 | 385,380.15 |
| 41 | 306,925.47 | 14,499.46 | 647.78 | 29.78 | 1.42 | 322,103.91 |
| 42 | 361,482.90 | 23,879.83 | 1014.93 | 43.14 | 1.87 | 386,422.67 |
| 43 | 305,436.54 | 14,327.37 | 636.33 | 28.48 | 1.26 | 320,429.99 |
| 44 | 336,065.58 | 22,602.88 | 982.95 | 41.47 | 1.73 | 359,694.61 |
| 45 | 336,065.58 | 20,352.15 | 1092.66 | 52.75 | 2.29 | 357,565.42 |
| 46 | 336,065.57 | 22,073.91 | 1186.35 | 60.06 | 2.79 | 359,388.68 |
| 47 | 336,065.58 | 20,831.33 | 1098.05 | 55.78 | 2.75 | 358,053.48 |
| 48 | 336,065.57 | 21,776.81 | 1170.42 | 59.85 | 2.93 | 359,075.58 |
| 49 | 336,065.58 | 21,080.38 | 1119.25 | 57.08 | 2.81 | 358,325.10 |
| 50 | 336,065.57 | 21,607.56 | 1159.90 | 59.43 | 2.92 | 358,895.38 |
| 51 | 336,065.58 | 21,215.80 | 1130.39 | 57.78 | 2.85 | 358,472.40 |
| 52 | 336,065.57 | 21,511.26 | 1153.23 | 59.10 | 2.91 | 358,792.07 |
| 53 | 336,065.58 | 21,290.75 | 1136.44 | 58.15 | 2.86 | 358,553.78 |
| 54 | 336,065.57 | 21,456.68 | 1149.24 | 58.89 | 2.90 | 358,733.28 |
| 55 | 336,065.58 | 21,332.57 | 1139.75 | 58.35 | 2.87 | 358,599.12 |
| 56 | 336,065.57 | 21,425.82 | 1146.94 | 58.76 | 2.89 | 358,699.98 |
| 57 | 336,065.58 | 21,355.99 | 1141.58 | 58.46 | 2.88 | 358,624.48 |
| 58 | 336,065.57 | 21,408.42 | 1145.62 | 58.69 | 2.89 | 358,681.18 |
| 59 | 336,065.58 | 21,369.13 | 1142.60 | 58.51 | 2.88 | 358,638.71 |
| 60 | 336,065.57 | 21,398.61 | 1144.87 | 58.65 | 2.89 | 358,670.58 |
| 61 | 336,065.58 | 21,376.51 | 1143.17 | 58.55 | 2.88 | 358,646.70 |
| 62 | 336,065.56 | 21,393.09 | 1144.45 | 58.62 | 2.89 | 358,664.61 |
| 63 | 336,065.58 | 21,380.66 | 1143.49 | 58.57 | 2.88 | 358,651.19 |
| 64 | 336,065.56 | 21,389.98 | 1144.21 | 58.61 | 2.89 | 358,661.25 |
| 65 | 336,065.58 | 21,383.00 | 1143.67 | 58.58 | 2.88 | 358,653.71 |
| 66 | 336,065.56 | 21,388.24 | 1144.07 | 58.60 | 2.89 | 358,659.36 |
| 67 | 336,065.59 | 21,384.31 | 1143.77 | 58.58 | 2.88 | 358,655.13 |
| 68 | 336,065.56 | 21,387.25 | 1144.00 | 58.60 | 2.89 | 358,658.29 |
| 69 | 336,065.59 | 21,385.05 | 1143.83 | 58.59 | 2.88 | 358,655.93 |
| 70 | 336,065.56 | 21,386.70 | 1143.96 | 58.59 | 2.89 | 358,657.69 |

By Tables A2 and A3 (for population sizes up to time step $t = 43$ and after time step $t = 44$, respectively), the logistic predator dynamics of each level $n$ of the simulated trophic pyramid are presented in Figures A6–A10 below:

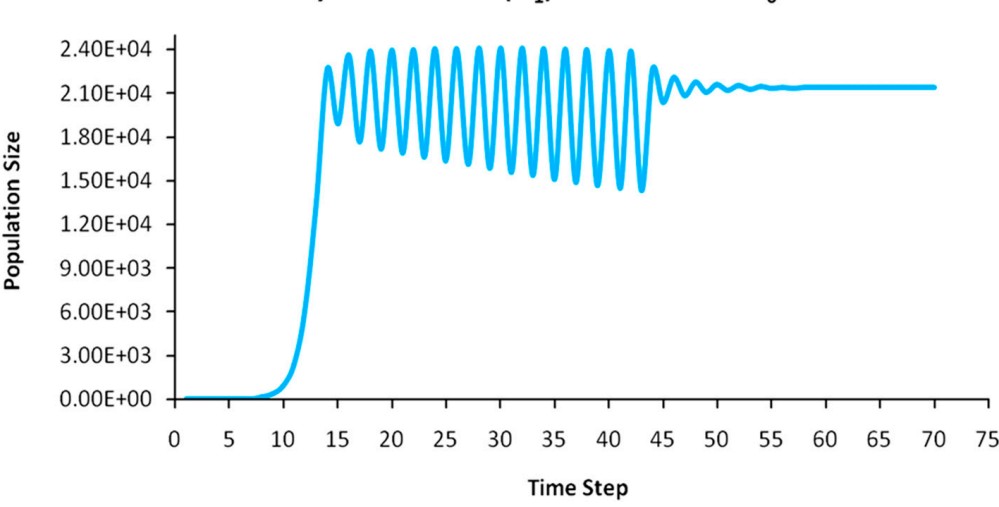

**Figure A6.** Growth dynamics of population $N_0$ with population control applied to it.

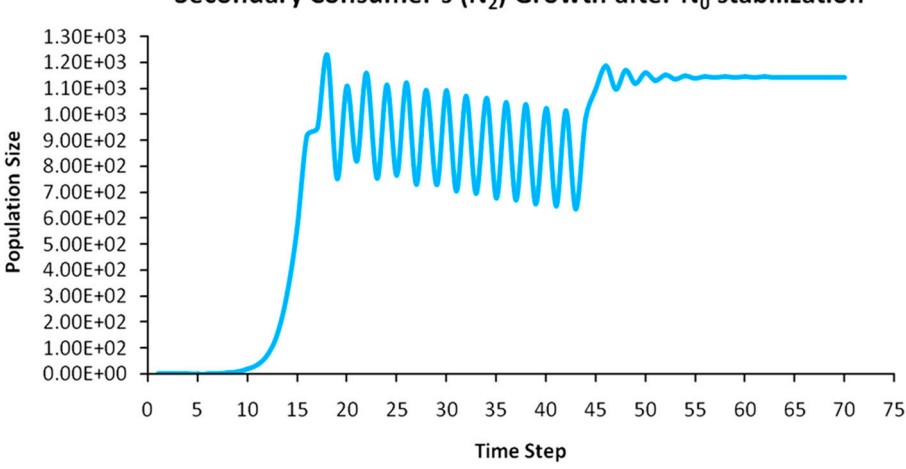

**Figure A7.** Growth dynamics of population $N_1$ with population control applied to $N_0$.

**Figure A8.** Growth dynamics of population $N_2$ with population control applied to $N_0$.

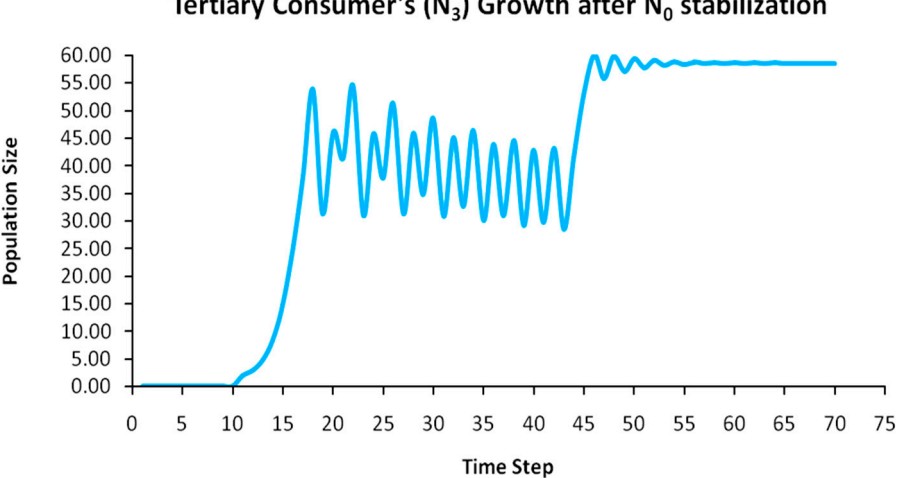

**Figure A9.** Growth dynamics of population $N_3$ with population control applied to $N_0$.

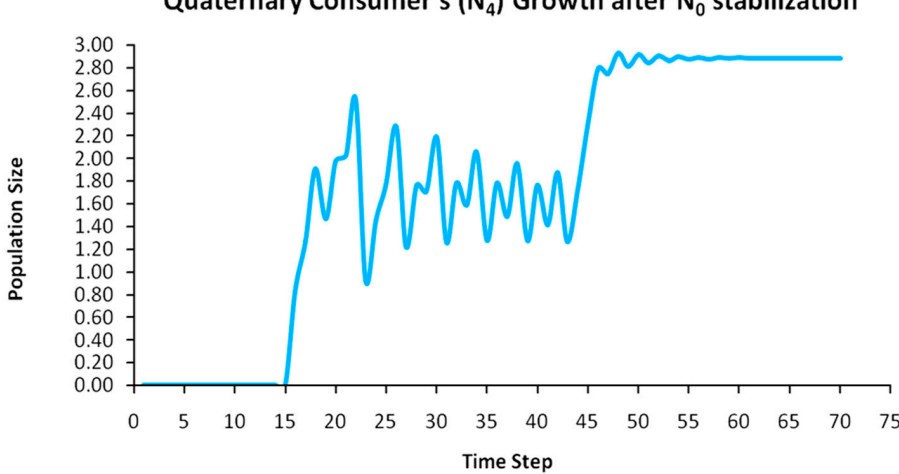

**Figure A10.** Growth dynamics of population $N_5$ with population control applied to $N_0$.

*Appendix A.3. Trophic Pyramid Stabilization Effects*

In this section, we present the quantitative framework of the three descriptive statistics measures presented in Tables A4 and A5 *before* and *after* the *key population's* stabilization and the deriving stabilization of each level $n > 0$ of the trophic pyramid. For increasing the reliability of the descriptive statistics' calculations, only specific simulation time frames were considered (i.e., time steps with non-stationary monotonic population growth were ignored for every pyramid level). In this context, all calculations are made from the first oscillation of each population (see grey-scaled values in Table A2), as also suggested in Equation (21) for estimating the values of the *adjusted Shannon entropy* index $H(N)_{ADJ}$. Hence, the first oscillation is considered to occur at time step $t^*$, where the size of a population is followed by its first reduction after constant growth. According to Table A2, this period is for $N_0$ at $t = 12$, for $N_1$ at $t = 14$ and for $N_2$, $N_3$ and $N_4$ at $t = 18$. In the first case, these are the time steps from which calculations are implemented until the last simulation time step ($t = 70$) for every separate pyramid level. In the second case, the calculations are performed only from time step $t = 44$ to time step $t = 70$. Specifically in the second case, the initial calculation at time step $t = 44$ is common for all pyramid levels, as they have all entered a phase of oscillations (see the grey-scaled values in Table A3).

As shown in Tables A4 and A5, calculations concern three (3) descriptive statistics measures: (a) the *weighted-average population* (=W-Average*N*), (b) the *population variance* (=Var*N*) and (c) the *population standard deviation* (=StDev*N*), as presented below.

Appendix A.3.1. Weighted-Average Population (W-Average*N*)

With each pyramid level having unequal weight on total biomass, the W-Average*N* calculations were considered necessary to measure each *n* level's real contribution to the pyramid's instability. The W-Average*N* was calculated for both time ranges. Therefore, population weights $W_{Nnt}$ are for each *n*-level species at each time step *t* equal to:

$$W_{Nn\sum_t t} = \frac{N_{nt}}{\sum_t N_t} \tag{A1}$$

Hence, the weighted population $N_{itW}$ for each *n*-level species at each time step *t* is:

$$N_{n\sum_t tW} = N_{n\sum_t t} \cdot W_{Nn\sum_t t} \tag{A2}$$

From Equation (A2), the W-Average*N* for each *n*-level species from the first oscillation is:

$$\hat{N}_n = \sum_{t=t^*}^{70} \frac{N_{n\sum_t tW}}{70 - t^*} \tag{A3}$$

The W-Average*N* for each *n*-level species from *t* = 44 is:

$$\hat{N}_n = \sum_{t=44}^{70} \frac{N_{n\sum_t tW}}{27} \tag{A4}$$

Appendix A.3.2. Population Variance (Var*N*)

As in Section A.3.1, the Var*N* for each *n*-level species from the first oscillation is:

$$S_n = \frac{\sum_{t=t^*}^{70} \left( N_{n\sum_t tW} - \hat{N}_n \right)^2}{70 - t^*} \tag{A5}$$

Respectively, the Var*N* for each *n*-level species from *t* = 44 is:

$$S_n = \frac{\sum_{t=44}^{70} \left( N_{n\sum_t tW} - \hat{N}_n \right)^2}{27} \tag{A6}$$

Appendix A.3.3. Population Standard Deviation (StDev*N*)

Finally, according to the formulation for the calculation of population variance, the population standard deviation $\sigma$ for each *n* species is for both cases:

$$\sigma_n = \sqrt{S_n} \tag{A7}$$

The W-Average*N* is fundamental as compatible to the *Eltonian Pyramid* concept according to Figures 3 and 4, Equation (7) and the *Herfindahl–Hirschman* index in Equation (30) and Table 4 for both empirical [34] and simulated diversity. In relation to the global empirical data, the simulation demonstrates low *vertical* diversity (i.e., across the trophic pyramid's levels), as suggested by the very high W-Average*N* value of Producers. This is a verification of Lindeman's rule on 10% energy efficiency and the consequent minimum energy requirements for biomass formation that pose an upper limit on Consumers. As discussed

in Section 4.1, with Producers' pivotal role as a metabolic hub of solar energy into secondary chemical energy stored in biomass, a *zero-sum game* is established for upper levels, where energy flow at one level occurs at the expense of another and at a cost of 90% spent on metabolic functions for structuring the biomass. Table A4 presents the absolute values of the three descriptive statistics' measures *before* $N_0$ stabilization for both periods $t^* \rightarrow t = 70$ and $t = 44 \rightarrow t = 70$ for each separate pyramid level, while Table A5 presents the respective results *after* $N_0$ stabilization.

**Table A4.** Calculations for all *n*-level populations and total trophic pyramid biomass ($\sum N$) of the three descriptive statistics' measures *before* population control is applied on Producers at $t = 44$.

| | Before Population $N_0$ Stabilization | | | | | |
| --- | --- | --- | --- | --- | --- | --- |
| | Changes (Absolute) of $N$ after 1ˢᵗ Oscillation | | | Changes (Absolute) of $N$ after $t = 44$ | | |
| | W-Average$N$ | Var$N$ | StDev$N$ | W-Average$N$ | Var$N$ | StDev$N$ |
| $N_0$ | $3.16 \times 10^5$ | $5.21 \times 10^8$ | $2.28 \times 10^4$ | $3.15 \times 10^5$ | $8.45 \times 10^8$ | $2.91 \times 10^4$ |
| $N_1$ | $1.09 \times 10^3$ | $1.59 \times 10^5$ | $3.99 \times 10^2$ | $1.02 \times 10^3$ | $1.71 \times 10^5$ | $4.13 \times 10^2$ |
| $N_2$ | $2.06$ | $6.30 \times 10^{-1}$ | $7.90 \times 10^{-1}$ | $1.80$ | $3.80 \times 10^{-1}$ | $6.20 \times 10^{-1}$ |
| $N_3$ | $3.00 \times 10^{-3}$ | $2.00 \times 10^{-6}$ | $1.40 \times 10^{-3}$ | $3.30 \times 10^{-3}$ | $8.00 \times 10^{-7}$ | $8.90 \times 10^{-4}$ |
| $N_4$ | $6.70 \times 10^{-6}$ | $7.00 \times 10^{-12}$ | $2.60 \times 10^{-6}$ | $6.40 \times 10^{-6}$ | $2.00 \times 10^{-12}$ | $1.20 \times 10^{-6}$ |
| $\sum N$ | $3.54 \times 10^5$ | $9.80 \times 10^8$ | $3.13 \times 10^4$ | $3.54 \times 10^5$ | $1.51 \times 10^9$ | $3.89 \times 10^4$ |

**Table A5.** Calculations for all *n*-level populations and total trophic pyramid biomass ($\sum N$) of the three descriptive statistics' measures *after* population control is applied on Producers at $t = 44$.

| | After Population $N_0$ Stabilization | | | | | |
| --- | --- | --- | --- | --- | --- | --- |
| | Changes (Absolute) of $N$ after 1st Oscillation | | | Changes (Absolute) of $N$ after $t = 44$ | | |
| | W-Average$N$ | Var$N$ | StDev$N$ | W-Average$N$ | Var$N$ | StDev$N$ |
| $N_0$ | $3.15 \times 10^5$ | $1.35 \times 10^8$ | $1.16 \times 10^4$ | $3.15 \times 10^5$ | $1.02 \times 10^5$ | $3.20 \times 10^2$ |
| $N_1$ | $1.21 \times 10^3$ | $8.03 \times 10^4$ | $2.83 \times 10^2$ | $1.28 \times 10^3$ | $1.85 \times 10^3$ | $4.30 \times 10^1$ |
| $N_2$ | $2.98$ | $7.80 \times 10^{-1}$ | $8.80 \times 10^{-1}$ | $3.60$ | $4.40 \times 10^{-2}$ | $2.10 \times 10^{-1}$ |
| $N_3$ | $6.90 \times 10^{-3}$ | $7.50 \times 10^{-6}$ | $2.70 \times 10^{-3}$ | $9.30 \times 10^{-3}$ | $1.00 \times 10^{-6}$ | $9.80 \times 10^{-4}$ |
| $N_4$ | $1.50 \times 10^{-5}$ | $1.00 \times 10^{-10}$ | $7.60 \times 10^{-6}$ | $2.20 \times 10^{-5}$ | $1.00 \times 10^{-11}$ | $3.20 \times 10^{-6}$ |
| $\sum N$ | $3.57 \times 10^5$ | $2.90 \times 10^8$ | $1.70 \times 10^4$ | $3.59 \times 10^5$ | $1.33 \times 10^5$ | $3.64 \times 10^2$ |

The percent (%) changes of each descriptive statistics measure presented in Tables A4 and A5, are shown in Table A6 of the next appendix section.

*Appendix A.4. Interpretation of Simulation Results*

We generated a Monte Carlo simulation by 100 iterations of a five-level trophic pyramid with constant parameter values (see Table A1). Iterations give oscillating populations for 100 random initial population values ($=N_{n0}$) for each of the pyramid's five levels and for 70 time steps. The simulation is a numerical validation of our theoretical conclusions. From these 100 iterations, we then calculated the average population value *for each time step* and *for each trophic pyramid level*, which we used for the three descriptive statistics measures as presented in Tables A4 and A5 that comprise the basis of ecological diversity indices. To demonstrate the trophic pyramid's stabilization effects in a straightforward way, we assume *Producers* to be the *key population* at the pyramid's base where population control is applied. We further assume five trophic levels ($0 \rightarrow 4$), as the maximum number of levels according to Lindeman [7]. As Lindeman argued, trophic pyramids may be expressed as a function of Producers' size, a view also verified by data [34] in Figure 4 that Equation (7) also follows. Specifically, Producers hold a significant share in conditioning the global climate [55], as the metabolism and flow of numerous chemical compounds in the lithosphere

and atmosphere (e.g., GHGs) highly depend on primary productivity in the biosphere that in turn depends on plants' biomass [6,51–55].

**Table A6.** Stabilization effects as percent changes for all *n*-level populations as well as for the total ecosystem biomass ($\sum N$) for 3 descriptive statistics measures; (a) *weighted average*; (b) *variance* and (c) *standard deviation*.

| | Producers' Population $N_0$ Stabilization Effects Per Trophic Pyramid Level *n* | | | | | |
| --- | --- | --- | --- | --- | --- | --- |
| | **% Changes of *N* after 1ˢᵗ Oscillation** | | | **% Changes of *N* after *t* = 44** | | |
| | **W-Average*N*** | **Var*N*** | **StDev*N*** | **W-Average*N*** | **Var*N*** | **StDev*N*** |
| $N_0$ | −0.07% | −74.16% | −49.17% | −0.14% | −99.99% | −98.90% |
| $N_1$ | 11.03% | −49.51% | −28.94% | 24.67% | −98.92% | −89.61% |
| $N_2$ | 44.19% | 24.69% | 11.67% | 99.20% | −88.56% | −66.17% |
| $N_3$ | 77.89% | 265.78% | 91.25% | 180.04% | 19.37% | 9.26% |
| $N_4$ | 129.99% | 760.72% | 193.38% | 248.59% | 561.33% | 157.16% |
| $\sum N$ | 0.67% | −70.35% | −45.55% | 1.46% | −99.99% | −99.06% |

Furthermore, at national and regional scales, even the temporary reduction in plant biomass due to anthropogenic or natural events, such as wildfires [78], deprives an ecosystem of its full ability to metabolize chemical compounds. Within a biophysical context, a significant contributor to intense wildfires is the displacement of grazing animals from wild lands, leaving excessive amounts of dead dry biomass in the ground, increasing the risk of combustion. Hence, although in terms of biomass, Producers dominate (as shown in Figure 4), the contribution of upper pyramid levels to the conservation of plant biomass via their biological functions is of vital importance. The depletion of both *horizontal* (within a trophic level) and *vertical* (across trophic levels) biodiversity intensifies limiting factors [54], as suggested by Figure 1c. With lack of biomass, most of the nutrients practically remain unutilized and embody a high risk of being permanently removed through water flows. At the economic level, the reduction in metabolic ability results in the loss of ecosystem services' value and related activities (e.g., fishing, timber, medicine).

In our simulation, the oscillating Producers' population causes via a linear topology respective oscillations to all upper trophic pyramid levels as well, resulting in an oscillating total ecosystem biomass. However, it is important to denote that our example is one of the simplest possible cases, having set $r_0 \cdot a_0 > 3$ for Producers that gives an oscillating population at the pyramid's foundation (as *K* is assumed constant for Producers). Hence, all oscillations in all upper levels occur mainly due to the variable carrying capacity in the lower ones and secondarily due to their $r_n \cdot c_n$ values that may give either stable or oscillating but convergent stable population sizes. We can assume other more complex cases, where oscillations in any Consumer population level are a composition of an oscillating carrying capacity (at the lower-level prey population) and a high intrinsic growth rate. Such a case in its most complex form would concern all levels of the trophic pyramid so that it would *primarily* require decomposition metrics to identify the statistical importance of each factor, and *secondarily*, population control at each separate pyramid level. These cases are the most challenging from an ecological engineering practice standpoint and the most expensive from an economic standpoint due to the *compartmentalization* and *statistical independence* of trophic pyramid levels. Such aspects of logistic predator models could be the theoretical benchmark for the development of *trophic pyramid typologies* by applying *information entropy* metrics [60] to identify synergistic adaptation patterns via *mutual information*. In Section 4.1 we discuss this aspect; however, its thorough quantitative analysis escapes our current scope and is left for future work.

Table A6 shows the percent (%) changes in three descriptive statistics of the simulation for every trophic pyramid level, as well as for total pyramid biomass ($\sum N_n$) *before* and *after* the Producer's population control, for a total period of 70 time steps. As mentioned, each Consumer level shares very high *mutual information* with Producers, meaning that their

population sizes highly depend on the fundamental population so that each instability or stabilization effect is *inductively* transferred to all upper levels with high efficiency (without information loss). Specifically, Table A6 presents the results: (a) if no optimal population control is applied to Producers and (b) after optimal population control is applied at time step $t = 44$, as shown in Tables A4 and A5. Percent population changes for all trophic pyramid levels were calculated by *weighted average* (W-Average$N$), *variance* (Var$N$) and *standard deviation* (StDev$N$) to demonstrate the effect of targeted population control. Calculations were implemented for two intervals of the total 70-time-step period: (1) from the time step recording the first oscillation of each population, which is the first time step $t^*$ at which the population size is lower than the previous ($N_{nt} < N_{nt-1}$), hence excluding the net (nonstationary) growth part at the initial steps that would generate bias in the measured sample; and (2) from the time step $t = 44$ where population control is applied to Producers, as shown in the second part of Table A6 (see Tables A2–A5 for details).

For each trophic level, we assumed a positive initial population $N_0$. For all Consumer levels, an additional condition by Equation (13) was added, so that this initial population size at any level $n$ manifests and starts growing only if the $n - 1$ level biomass is sufficient to sustain it by the minimum energy requirements of Table A1. Thus, as shown in Tables A2 and A3 for both measured periods, population sizes of Consumer levels $1{\rightarrow}4$ are zero at the initial time steps. Biophysically, this can be interpreted as an initial inexistence of a trophic pyramid and its formation only after a critical biomass size of Producers has been formed so that other species (from herbivores to top predators) gradually migrate due to resource availability.

Producers' population stabilization affects significantly all descriptive statistics, with the most significant observed for variance and standard deviation in regard to oscillation intensity. Variances and standard deviations were calculated for weighted averages, as irrespective of the oscillation intensity of each separate pyramid level, their weights to the overall ecosystem biomass are highly unequal. For instance, $N_0$ comprises 94–95% of all ecosystem biomass—near the empirical fraction presented in Figure 4—while $N_0$ and $N_1$ combined comprise ~99.8%, and $N_2, N_3$ and $N_4$ combined comprise only the residual 0.2%. From Tables A4 and A5, the W-Average$N$ for $N_0$ is slightly lower by 0.07% from the first oscillation and by 0.14% from $t = 44$. However, this reduction was overcompensated by the population increase in the upper trophic levels, leading to a small overall increase in the pyramid's biomass by 0.67% from the first oscillation ($t^*$) and by 1.46% from $t = 44$. What is also remarkable is the decrease in Var$N$ and StDev$N$ for both periods. Decreases in Var$N$ and StDev$N$ for $N_0$ and $N_1$ have significant weight on the pyramid's total biomass Var$N$ and StDev$N$. $N_2$ shows particular behavior with an increase in Var$N$ and StDev$N$ from the first oscillation but significant decrease from $t = 44$. This is mostly due to the significantly higher W-Average$N$ after the stabilization that mainly affects all time steps after the first oscillation (see Tables A4 and A5) affecting the Var$N_2$ and StDev$N_2$ after $t^*$. In contrast, Var$N_2$ and StDev$N_2$ after $t = 44$ show notable decrease. As after stabilization at $t = 44$, $N_3$ and $N_4$ stabilize at sizes above their oscillation limits, their Var$N$ and StDev$N$ increase for both periods.

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
