# Peer review of "Energy, Trophic Dynamics and Ecological Discounting"

_land, doi:10.3390/land12101928_

Round 1
Reviewer 1 Report
The article addresses a subject of undeniable interest; however, in light of the title that heads it, it incurs a certain thematic insufficiency. The generality of the title needs to provide a clear apprehension of the scope and focus of the content. Regarding the introduction, although my expertise is focused on ecosystem services and not energy, it is clear that the treatment given to ecosystem services in the introduction is limited. Restricted use of references is observed, some of which are good (such as Costanza), although I consider a more exhaustive and updated literature review necessary. The omission of references to established frameworks of ecosystem services, such as The Common International Classification of Ecosystem Services (CICES), could suggest a superficial understanding of the scope of ecosystem services. In the methodological section, given the broad scope of the objectives, it is difficult to discern whether the proposed methodology is sufficient or appropriate; This same reservation could be extended to the results and conclusions, which leads to reflection on the solidity of these aspects. Finally, the proposal is innovative, and I recommend doing a literature review as this is a very general topic.
Reviewer 2 Report
The paper looks very rigorous, discussing an important issue - energy - from a quantitative perspective, and making use of mathematical formulas for the general development of this field. This must be applauded.
I would suggest however to improve the paper by:
- shortening the abstract, which is too long. please check the guidelines of MDPI
- positioning the paper within the relevant literature. For instance, I would suggest reviewing and including the latest work of Prof Benjamin Schuetze, who has been working on this topic of energy from a qualitative perspective. See in particular: Schuetze, Benjamin "The geopolitical economy of an undermined energy transition: The case of Jordan." Energy Policy 180 (2023): 113655.
Moreover, maybe the literature on the WEF nexus may also be useful to frame your paper and work?
Author Response
Please, see attachment.

Reviewer 3 Report
Ms. Ref. No.: land-2561653-peer-review-v1
Title: Energy, trophic dynamics and ecosystem services
Article Type: Article
I found this study interesting. The present work focuses on the biotic part and specifically on the impact of a trophic pyramid’s endogenous stability on the longterm generation of ecosystem services deriving from the total biomass embodied in the species’ populations forming the pyramid. The study conducted here could give new impulses for the scientific community. The manuscript requires modifications before it is suitable for publication (see specific comments).
Specific comments
Abstract
Generally, the authors should rewrite the abstract more clearly. I suggest the authors to write a sentence of background before the aims and to rewrite better aims, methods, major findings and conclusion more clearly
Keywords
Replace two or more appropriate key words.
1. Introduction
Introduction should state more clearly the reason for doing the work! It is long. It reads rather like a review than a short introduction to topic.
2. Materials and Methods
I am lost!! I think that reads rather like a review than a short Materials and Methods to topic of the manuscript. Please merge, rewrite shortly and clearly the statements!!
3. Results
I am lost!! The writing style of this section is insufficient. Please rewrite shortly and clearly the most places with main results and figures. Are needed all figures? (supplementary material)
4. Discussions
I am lost!! The authors following the results need to rewrite shorter subchapters with resent references.
5. Conclusion
I suggest the authors to write again the main conclusions clearly according the aims of the study!
Need to add future prospects??
Extensive editing of English language required
Author Response
Please, see attachment.

Round 2
Reviewer 1 Report
Dear Authors.
Compared to the previous version, I have seen a substantial improvement in the work.
I have no further comments on my part.
Reviewer 2 Report
The paper is improved, as the authors have incorporated the comments received by the different reviewers. It makes it more comprehensive. My suggestion for improving the introduction is to situate it within the current dynamics, in particular looking at the energy side could be useful to also mention the implications of the European Green Deal, as emphasised by the latest work of Serena Sandri (Italian-german scholar) in the journal "Mediterranean Politics".
Moreover, when discussing the work of Schuetze, maybe also reference to the paper of Serena Sandri published in Sustainability on energy policies in Jordan could be a useful addition.
Once those two edits are incorporated, I would be happy to recommend acceptance of this paper for publication.
The paper on the green deal provides insights on how a regional policy can shape trade, sustainability, and exchanges dynamics including on energy and related policies. The paper on national energy policies is relevant as it discusses the different dynamics that contribute to shape and inform energy policies in countries. I would also suggest exploring the work of Benjamin Schuetze and his new project on energy policies in North Africa and the interface of science and policy. The work of the istituto affari internazionali on the topic is also worth considering.
Reviewer 3 Report
Ms. Ref. No.: land-2561653
Title: Energy, trophic dynamics and ecosystem service
I found this study interesting. In general, the paper now has improved. I think it is suitable for publication with minor language corrections.
Minor editing of English language required.
